# Baculoviruses remodel the cytoskeleton of insect hemocytes to breach the host basal lamina
Ryuhei Kokusho ⓘ ✉ & Susumu Katsuma ⓘ ✉

Many pathogens and endosymbionts hijack the host's cytoskeleton for efficient propagation and transfer within or between host cells. Once released into the host's circulatory system, however, they have to confront structural barriers without utilizing host cell functions. Many insect viruses and insect-borne viruses can re-enter from the hemolymph into insect tissues despite the barrier of the basal lamina (BL), but the molecular mechanism remains unclear in many cases. Here, we demonstrate that Bombyx mori nucleopolyhedrovirus (BmNPV) remodels host hemocytes to breach the BL. We found that the viral membrane protein actin rearrangement-inducing factor 1 (ARIF-1) induces filopodia-like protrusions and invadosome-like structures in hemocytes, which play a critical role in attaching to the tissue surface, penetrating the tracheal BL and thus facilitating the transport of viral nucleocapsids into host tissues. Our findings clearly show the role of hemocyte infection in viral systemic spread and its molecular basis.

After primary infection, the ability of pathogens and endosymbionts to spread within the host body (secondary infection) is critical for maximizing their propagation and dispersal. Viruses employ various strategies to facilitate this spread, including the manipulation of the host cytoskeletal system to transfer to neighboring cells or infect adjacent tissues[1–6]. In contrast, once viruses are released into the host's circulatory system, they can no longer utilize the host's cellular functions. Thus, they often require specific mechanism(s) for re-entry into host tissues or organs. Typically, the physical structures present at the interface between the host's tissues and circulatory system, such as the blood-brain barrier, pose a significant barrier to pathogenic invasion[7,8]. In invertebrates with an open circulatory system, most tissues are covered with the basal lamina (BL), a dense fibrous layer of the extracellular matrix (ECM). The BL layer acts as a defense system, shielding tissues against direct pathogenic invasion from the hemolymph[9]. Some insect viruses, arthropod-borne mammalian viruses, and insect vector-mediated plant viruses are capable of passing through the BL to escape from the midgut and re-enter into host tissues; hence, these viruses can move to the insect tissues appropriate for efficient viral propagation or transmission[1,7]. However, the molecular mechanism underlying how these viruses can re-enter host tissues from the hemolymph remains unclear in many cases.

Baculoviruses are one of the best-studied insect-associated viruses regarding the mechanism of overcoming the host BL barrier and resulting systemic infection. Baculoviruses are entomopathogenic double-stranded DNA viruses, which are further divided into four genera; *Alphabaculovirus* (lepidopteran-specific nucleopolyhedroviruses (NPVs)), *Betabaculovirus* (lepidopteran-specific granuloviruses), *Gammabaculovirus* (hymenopteran-specific NPVs), and *Deltabaculovirus* (dipteran-specific NPVs). Baculoviruses produce two types of virions, occlusion-derived virus (ODV) and budded virus (BV); ODVs are occluded within an occlusion body (OB) that protects and transmits ODVs from insect to insect via oral infection, whereas BVs are required for cell-to-cell transmission and are involved in the spread of virus infection within an infected host. Initially, ODV infection occurs in the midgut under natural conditions (Supplementary Fig. S1A, [i–ii]). Next, most baculoviruses cross the midgut BL to establish infection in other tissues. However, their rod-shaped nucleocapsids are considered too large (200–400 nm in length and 30–50 nm in diameter[10,11]) to pass through the tiny pores of the BL (15 nm in diameter[12]). Granados and Lawler (1981) observed *Autographa californica* multiple nucleopolyhedrovirus (AcMNPV, *Alphabaculovirus aucalifornicae*) virions pushing the basal side of the *Trichoplusia ni* midgut epithelial cells and released into the hemolymph at the very early stage of oral infection[13], suggesting direct crossing of the midgut BL barrier by an unknown mechanism (Supplementary Fig. S1A, [iii-a])[14]. Alternatively, many baculoviruses are considered to utilize the host trachea as an escape route from the midgut (Supplementary Fig. S1A, [iii-b]), based on the studies of alphabaculoviruses[15,16]. At the interfaces between the trachea and other tissues, tracheal terminal cells (tracheoblasts) penetrate the BL to oxygenate cells in the surrounding tissues

Department of Agricultural and Environmental Biology, Graduate School of Agricultural and Life Sciences, The University of Tokyo, Tokyo, Japan.
✉e-mail: rkokusho@g.ecc.u-tokyo.ac.jp; skatsuma@g.ecc.u-tokyo.ac.jp

efficiently. In this midgut escape route, baculoviruses infect tracheoblasts after midgut infection to bypass the BL barrier, leading to the progression of tracheal epithelial cell infection toward distant tissues (Supplementary Fig. S1A, [iv-a]). Presumably, direct BV release from tracheoblasts into the hemolymph also facilitates systemic infection (Supplementary Fig. S1A, [iv-b]).

In contrast to many reports on the midgut escape of baculoviruses, fewer studies have been conducted on the viral re-entry from the host hemocoel into host tissues. Baculoviruses have to cross the BL barrier of host tissues for re-entry. Exceptionally, virions can directly infect hemocytes that do not have the BL barrier (Supplementary Fig. S1A, [v-b]). It is commonly accepted that BVs in the hemolymph next infect tracheoblasts on the surface of distant tissues to bypass the BL, and then the infection spreads inside the tissues (Supplementary Fig. S1A, [v-a, vi])[9,14–16]. Alternatively, Keddie et al. [17] reported virus-infected hemocytes (insect blood cells) attaching to the epidermis at the early stage of infection[17], implying the importance of hemocytes for the systemic spread of infection (Supplementary Fig. S1A, [vii]). However, because of the existence of the BL barrier, whether this attaching hemocytes is crucial for the spread of infection inside BL-coated tissues remains unknown. In this study, we discovered that the membrane protein ARIF-1 (actin rearrangement-inducing factor 1) of Bombyx mori nucleopolyhedrovirus (BmNPV, *Alphabaculovirus bomori*) modifies the actin cytoskeleton of host hemocytes and enables them to breach the BL barrier, establishing a highly efficient systemic infection.

## Results

### BmNPV can directly establish infection from the hemolymph at the central tracheal region despite the BL barrier

To examine the systemic infection route of BmNPV, we injected BVs directly into the larval hemocoel of the domesticated silkworm *Bombyx mori*. This infection method was employed to evaluate the route of direct infection from the hemolymph by minimizing the hemolymph-independent systemic infection via the tracheal route (Supplementary Fig. S1B). We used a GFP-expressing recombinant BmNPV (hspGFP) to track the spread of infection within tissues every 12 h after infection. Immunofluorescence staining revealed that GFP expression was initially observed in the trachea at 24 h post-infection (hpi), and by 60 hpi, the infection expanded to a larger area (Fig. 1A, upper panels). In contrast to the conventional understanding of systemic infection, we made a noteworthy discovery: the infection did not solely originate from tracheoblasts but also from the central tracheal region (near branching points of thick tracheal branches in many cases) that are presumably covered by the BL and have no tracheoblasts (Fig. 1B, upper panels).

Next, we examined whether this infection route via the central tracheal region is related to the BmNPV *arif-1* gene. ARIF-1 is a three-transmembrane protein that causes actin rearrangement during the early stages of infection (Supplementary Fig. S2A)[18]. *arif-1* is not a baculovirus core gene, but its homologs are present in most alphabaculovirus genomes[18,19]. ARIF-1 is essential for the formation of invadosomes in lepidopteran insect cultured cells[20]. In some mammalian cells and tumor cells, invadosomes are known as cellular structures that make close contact with the ECM and may destroy it for cell invasion[21,22]. While *arif-1* deletion has little effect on viral propagation in cultured cells[23,24], our earlier investigation on the temporal change of viral gene expression in host tissues revealed that *arif-1*-deleted BmNPV showed significantly delayed systemic spread of infection in *B. mori* larvae[23]. These studies suggest that ARIF-1 and ARIF-1-derived actin remodeling play an important role in the systemic spread of infection, although the precise histology and molecular mechanisms remain unknown. We noticed a significant delay in the spread of tracheal infection when we inoculated the *arif-1*-deleted variant of hspGFP (hspGFP-ARIFD) (Fig. 1A, lower panels). Notably, at 24 hpi, infection through the central tracheal region was 66.0-fold fewer in hspGFP-ARIFD-infected trachea (Fig. 1B, lower panels; Fig. 1C). At 48 hpi, hspGFP-ARIFD infection seemed to spread from the terminal tracheal region (Fig. 1D); this is probably caused by the conventional infection route through tracheoblasts. These results

suggest that ARIF-1 is a critical element in the alphabaculovirus infection through the central tracheal region, substantially increasing the efficiency of infection spread. Hereafter, we termed the conventional infection route through tracheoblasts as the "evasive" route and the new route originating from the central tracheal region (covered by the BL barrier) as the "invasive" route.

### ARIF-1 facilitates the generation of infection foci in the fat body

Next, we examined the spread of infection in the fat body. The fat body is considered a major destination for baculovirus systemic infection because this tissue is most abundant in the insect body and thus can be the primary source of viral OB production in an individual host. As shown in Fig. 2A, both hspGFP and hspGFP-ARIFD showed infection foci in the fat body at 24 hpi, many of which seemed to be adjacent to tiny tracheal branches. However, the number and the size of infection foci were different between these two viruses; the number of infection foci was 6.11-fold fewer in hspGFP-ARIFD compared with that of hspGFP (Fig. 2C), suggesting that ARIF-1 facilitates infection foci generation in the fat body or fat body-associated tracheal branches. Many large multicellular infection foci were observed in the hspGFP-infected fat body in addition to small single-cellular infection foci (Fig. 2A, upper panels), whereas the majority of infection foci appeared to be small and single-cellular in the hspGFP-ARIFD-infected fat body (Fig. 2A, lower panels). hspGFP showed two peaks in the size distribution of infection foci ($10^{2.2–2.4}$ μm$^2$ and $10^{3.4–3.6}$ μm$^2$), whereas hspGFP-ARIFD showed only one peak ($10^{2.2–2.4}$ μm$^2$) (Fig. 2D). These two peaks may be explained by two different infection routes; (i) smaller infection foci ($10^{2.2–2.4}$ μm$^2$) by the evasive infection via fat body-associated tracheoblasts, which does not require ARIF-1, and (ii) larger infection foci ($10^{3.4–3.6}$ μm$^2$) by invasive infection, which overcomes the BL barrier of the fat body or fat body-associated trachea by ARIF-1. The proportion of infection foci greater than 1000 μm$^3$ was over 7-fold higher in hspGFP-infected fat bodies than those infected with hspGFP-ARIFD (Fig. 2E), demonstrating the importance of ARIF-1 to form larger infection foci. This hypothesis is further supported by the observation at later stages of infection. The infection of hspGFP-ARIFD spread along the tracheal epithelia or showed patchy infection in the fat body at 36 hpi (Fig. 2B); this spreading pattern was different from large infection foci of hspGFP in the fat body that expands in a circular pattern around the tracheal branch points (Fig. 2A, B), suggesting that ARIF-1-mediated large infection foci are due to the different mechanism from the tracheoblast-mediated spread of infection. It was difficult to precisely distinguish by which route each infection focus was generated in the fat body because many tiny tracheal branches extend on the fat body. Therefore, in the following sections, we focused on examining the mechanism of the invasive route in the trachea.

### BmNPV-infected hemocytes attach to the invasive infection foci in the trachea

During microscopic observation, we found that cells with smaller nuclei than tracheal epithelial cells attached to the infection foci of the invasive route (Fig. 3A). Most of these attached cells expressed the Hemocytin protein (Fig. 3B), which is known to be exocytosed from granulocytes (insect blood cells for cellular immunity that account for 50–80% of total hemocytes in *B. mori* larvae[25]). We counted the number of attached hemocytes and found a >20-fold increase in hspGFP than hspGFP-ARIFD and mock (Fig. 3C), indicating that ARIF-1 markedly facilitates hemocyte attachment. The mean ratio of GFP- and Hemocytin-positive hemocytes attached to the trachea were 88.2% and 89.9%, respectively (Fig. 3D). 78.1% expressed both GFP and Hemocytin, whereas 10.1% were GFP-positive but Hemocytin-negative (Fig. 3E). These results suggest that ARIF-1-mediated hemocyte attachment mainly occurs in granulocytes, and other hemocyte types also have this ability.

Some populations of insect hemocytes are intrinsically sessile, i.e., attached to tissue surfaces[26–30]. Given that the number of attached hemocytes in the mock-infected larval trachea was markedly lower than that of hspGFP (Fig. 3C), hemocyte attachment in BmNPV-infected trachea appears to be

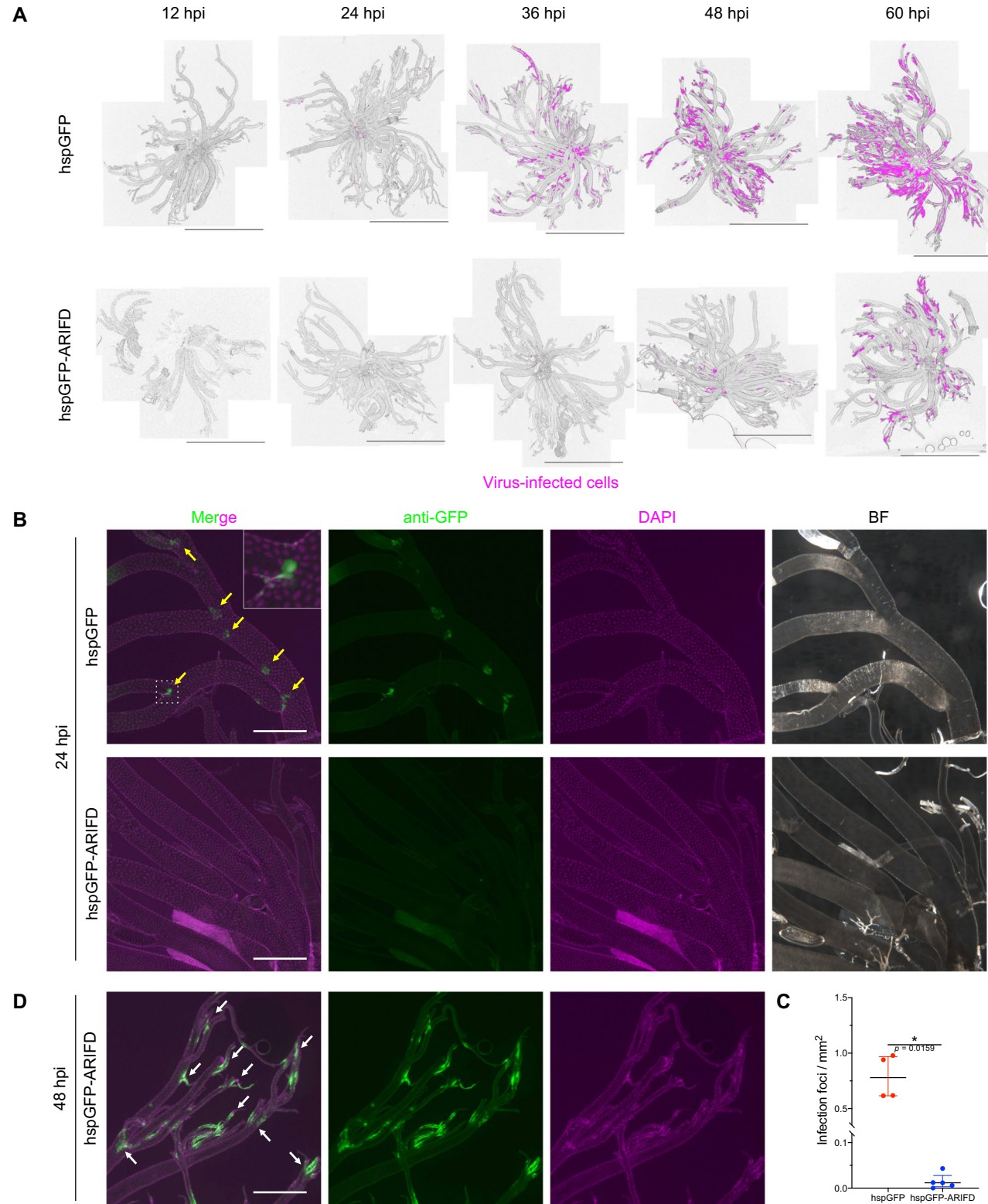

**Fig. 1 | BmNPV establishes infection at the central tracheal region by ARIF-1 despite the BL barrier.** Immunostained *B. mori* larval trachea infected with hspGFP or hspGFP-ARIFD were observed under zoom microscopy. **A** Spread of viral infection. Virus-infected cells (GFP-positive) of tracheal samples at designated time points are pseudocolored in magenta as the method described in the "Methods" section. Bar, 2.5 mm. **B** Infection foci (yellow arrows) at 24 hpi. The area of the white dashed square is enlarged, as shown in the inset. **C** The number of infection foci per area at 24 hpi. Data shown are median with interquartile range. *$p < 0.05$ by Mann–Whitney test (two-tailed). **D** hspGFP-ARIFD-infected tracheal tissues at 48 hpi. White arrows, virus-infected cells in the thin tracheal branches (presumably originating from tracheoblast infection). **B, D** anti-GFP (green), virus-infected cells; DAPI (magenta), dsDNA; BF, bright-field image. Bar, 500 μm.

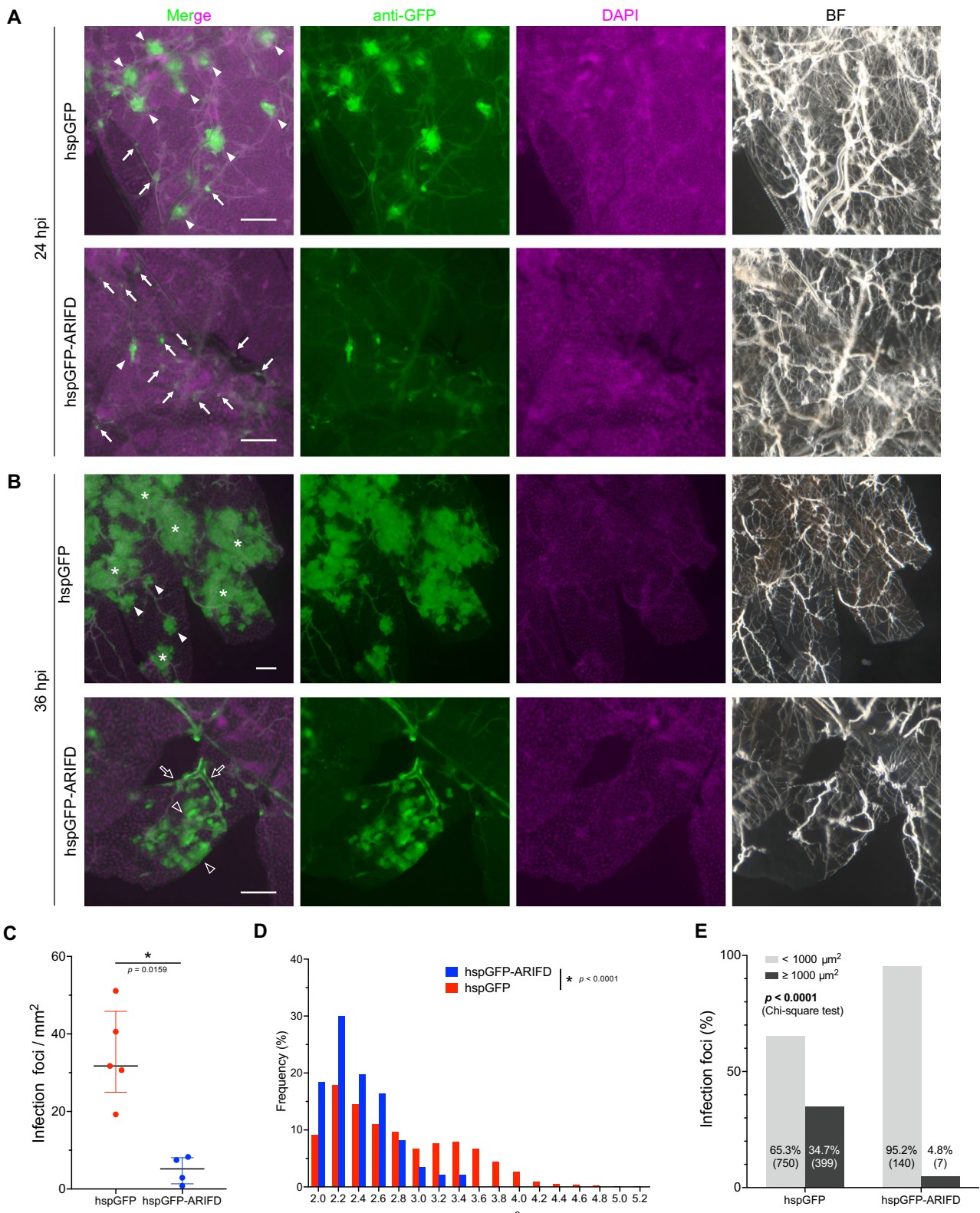

**Fig. 2 | ARIF-1 facilitates infection foci formation and expansion in the fat body.** Immunostained *B. mori* larval fat bodies infected with hspGFP or hspGFP-ARIFD were observed under zoom microscopy. Anti-GFP (green), virus-infected cells; DAPI (magenta), dsDNA; BF, bright-field image. Bar, 100 μm. **A** Infection foci at 24 hpi. Filled arrows and arrowheads indicate small and large infection foci, respectively. **B** Infection foci at 36 hpi. Filled arrowheads, large infection foci; asterisks, fat body infection foci expanding to a wider area than those at 24 hpi; outline arrowheads, fat body cell infection near the thin tracheal branches; outline arrows, tracheal cell infection along tracheal tubules. **C** The number of infection foci per area. Data shown are median with interquartile range. *$p < 0.05$, Mann–Whitney test (two-tailed). **D** Histogram of the areas of infection foci. hspGFP, $n = 1149$; hspGFP-ARIFD, $n = 147$. *$p < 0.001$ by Mann–Whitney test (two-tailed). **E** The percentage of infection foci over and under 1000 μm². The numbers in parentheses on the bar graph represent the number of infection foci observed. $p < 0.0001$ by Chi-square test.

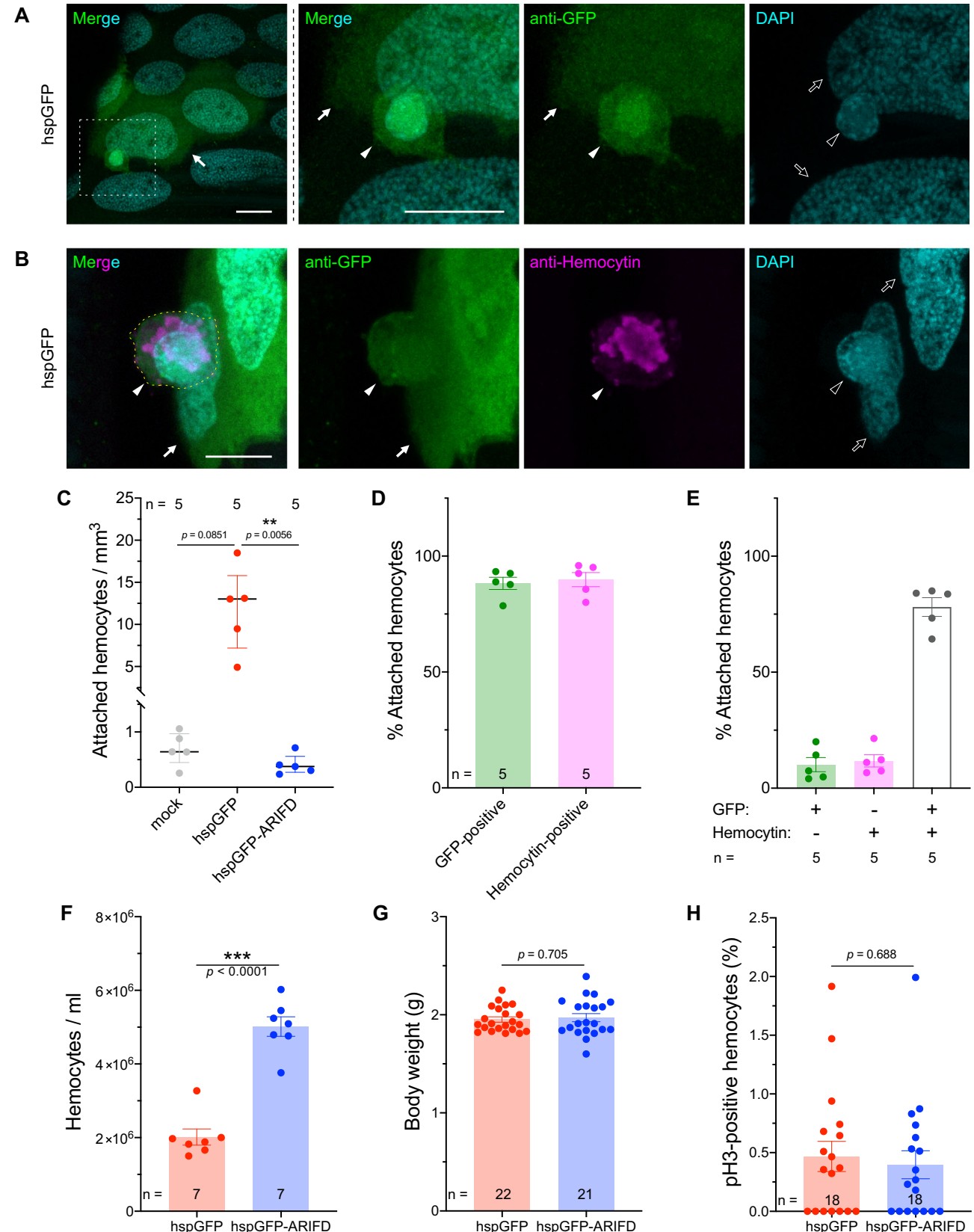

derived from circulating hemocytes rather than sessile ones. This was supported by the examination of circulating hemocytes; the number of hemocytes in the hemolymph was 2.49-fold higher in hspGFP-ARIFD-infected larvae than in hspGFP-infected larvae (Fig. 3F), whereas the body weight (Fig. 3G) and the percentage of mitotic hemocytes (Fig. 3H) were not substantially different between hspGFP and hspGFP-ARIFD. These findings suggest that around 60% of circulating hemocytes disappeared from the hemolymph after BmNPV infection, presumably due to the ARIF-1-

**Fig. 3 | BmNPV-infected hemocytes attach to the invasive infection foci in the trachea. A, B** Confocal microscopy Z-projection of the immunostained hspGFP-infected trachea at 24 hpi. Images representing attachment of hspGFP-infected hemocytes (filled arrowhead) on the infection foci (filled arrow) (**A**) and their Hemocytin expression (**B**) are shown. The area of the white dashed square is enlarged and shown in the right panels. Outline arrowheads and arrows, the nuclei of the attached hemocytes and tracheal epithelial cells, respectively. A yellow dashed line indicates the shape of the attached hemocyte. Anti-GFP antibody (green), virus-infected cells; anti-Hemocytin (magenta), a granulocyte marker; DAPI (cyan), dsDNA. Bar, 10 μm. **C–E** Characterization of hemocyte attachment. The number of GFP- or Hemocytin-positive hemocytes on the immunostained trachea from mock-, hspGFP-, and hspGFP-ARIFD-infected larvae were counted under zoom microscopy. **C** Density of hemocyte attachment (GFP- or Hemocytin-positive). Data shown are median with interquartile range (*n* = 5). **Adjusted *p*-value < 0.01, Kruskal–Wallis test with Dunn's multiple comparisons test with Bonferroni's correction. **D** Percentage of virus infection (GFP-positive) and granulocytes (Hemocytin-positive) in attached hemocytes in hspGFP. **E** The breakdown of each marker-positive/negative combination. Data shown in (**D**, **E**) are mean ± SEM (*n* = 5). **F–H** Analysis of circulating hemocytes. Larvae were mock-, hspGFP-, or hspGFP-ARIFD-infected and used for the following analysis. Data shown are mean ± SEM. ***p* < 0.001, unpaired *t*-test (two-tailed). **F** The number of circulating hemocytes in the hemolymph at 24 hpi. **G** Larval body weight at 24 hpi. **H** Percentage of hemocytes with phosphorylated Histone H3 (a mitosis marker) at 18 hpi.

induced attachment to the tissue surface. On the other hand, the possibility of lysis, clumping, or sequestration of hemocytes by AIRF-1 cannot be ruled out.

## BmNPV-infected hemocytes form filopodia-like protrusions and invadosome-like structures by ARIF-1

To elucidate the precise function of ARIF-1 in the attachment of virus-infected hemocytes to the tissue surface, we generated a recombinant BmNPV named ARIF-GFP, in which the native ARIF-1 was replaced with GFP-fused ARIF-1 (Supplementary Fig. S2A). In vitro experiments confirmed the expression and membrane localization of GFP-fused ARIF-1 in ARIF-GFP-infected cultured cells (Supplementary Fig. S2B–D), consistent with previous studies[20,24]. We also demonstrated that GFP-fused ARIF-1 was capable of enhancing systemic infection in *B. mori* larvae as the native ARIF-1 (Supplementary Fig. S2E, F). Using ARIF-GFP, we examined the virus-infected trachea and the attached hemocytes. As depicted in Fig. 4A, ARIF-GFP-infected hemocytes exhibited numerous filopodia-like protrusions enriched with ARIF-1, extending laterally along the tracheal surface. 3D reconstruction of confocal microscopic images revealed that some of them split into two or three branches (Fig. 4B). ARIF-1 showed colocalization with F-Actin in filopodia-like protrusions (Fig. 4C), which was further supported by Pearson's colocalization coefficient analysis (Fig. 4F). These results strongly suggest that the formation of filopodia-like protrusions is driven by actin cytoskeleton rearrangement. Additionally, we found another type of ARIF-1-rich protrusion—we named it the invadosome-like structure based on a prior in vitro study of AcMNPV ARIF-1[20]–, which extends to the basal side of attached hemocytes and appears to invade the BL (Fig. 4B). When the samples were gently crushed with a cover glass to rupture the attached hemocytes, we found invadosome-like aggregations of ARIF-1 on the tracheal surface, likely remnants of the attached hemocytes (Fig. 4D). Invadosome-like aggregations exhibited diversity in number and size, from fewer and larger dots to more and smaller ones, and sometimes showed fused structure (Supplementary Fig. S3). These invadosome-like structures were colocalized with F-Actin (Fig. 4E, F), similar to invadosomes induced by ARIF-1 in lepidopteran cultured cells[20]. These findings suggest that ARIF-1 has the ability to induce two types of cellular remodeling in the hemocytes of host insects: filopodia-like protrusions (slim cell protrusions extending laterally alongside the tissue surface) and invadosome-like structures (dot-like structures and its extension on the attaching basal surface of hemocytes). Given that 95.74% and 87.23% of the attached hemocytes formed filopodia-like protrusions and invadosome-like structures, respectively (Fig. 4G), these ARIF-1-derived structures are general features of BmNPV-infected hemocytes.

We performed an oral infection assay to examine whether hemocyte attachment, formation of ARIF-1-derived structures, and resulting infection spread in the BL-protected trachea occur in the case of the natural infection route. Zoom microscopic observation of immunostained tracheal samples revealed that hemocyte attachment was detected from 48 hpi (Supplementary Fig. S4A). Attached hemocytes formed filopodia-like protrusions and invadosome-like structures (Supplementary Fig. S4B). From 72 hpi, infection foci in the BL-surrounded area were detected (Supplementary Fig. S4A). These results suggest that

ARIF-1 enhances systemic spread of infection in oral infection in the same manner as intrahemocoelic infection.

Next, we observed the detailed structure of attached hemocytes by scanning electron microscopy (SEM). Consistent with the observations of immunostained samples (Fig. 3C), only a small number of hemocytes were found attached to the tracheal surface in mock- and hspGFP-ARIFD-infected larvae. In contrast, we found many attached hemocytes in hspGFP-infected trachea. Attached hemocytes from mock- and hspGFP-ARIFD-infected larvae kept their round shape and had only a small number of filopodial protrusions (Fig. 5A, C). In contrast, those from hspGFP-infected larvae extended their plasma membrane to the wide area of the tracheal surface to form mesh-like structures composed of thin cell protrusions (Fig. 5B, B'). Also, small vesicles were found attached near such mesh-like structures (Fig. 5B'). Such morphological changes appear similar to the disintegration and degranulation of lepidopteran granulocytes in the immune response[31], although we did not exclude the possibility of an artificial immune response due to dissection. These findings imply the ARIF-1's ability to induce unique structures that presumably contribute to cell attachment to the tissue surface.

## ARIF-1-derived structures work as migratory paths for viral nucleocapsids

We proposed that filopodia-like protrusions or invadosome-like structures in the invasive route cross the BL to deliver virus particles from virus-infected hemocytes into the BL-protected tissues. Immunostaining of collagen IV (ColIV), a component of the BL, revealed that some filopodia-like protrusions penetrated the tracheal BL (Fig. 6A), which was supported by the intensity plot of fluorescent signals (Fig. 6B). Similarly, invadosome-like structures were also found penetrating the BL barrier (Fig. 6C, Supplementary Fig. S5); in this case, intensity plots demonstrated that the GFP signal from the filopodia-like protrusion was located outside the ColIV signal (Fig. 6D), while the GFP signal from the invadosome-like structure overlapped with the ColIV signal and extended to the nucleus of the tracheal epithelial cell (Fig. 6E). The ratio of hemocytes with BL-invading filopodia-like protrusions and invadosome-like structures were 81.95% and 100.00%, respectively (Fig. 6F), indicating that BL invasion by these ARIF-1-derived structures is a typical phenomenon in BmNPV-infected attached hemocytes.

Next, we examined whether ARIF-1-derived structures can deliver viral nucleocapsids. Immunostaining of VP39, a viral major capsid protein, revealed the presence of nucleocapsids inside the filopodia-like protrusions and at their tips (Fig. 7A). DAPI signals were also detected in the same place with VP39 signals, although very weak and not completely colocalized (Fig. 7A). Similarly, VP39 signals were also observed inside and at the tip of invadosome-like structures (Fig. 7B). These results corroborate the nucleocapsid-delivering ability of ARIF-1-derived structures. The ratios of hemocytes with ARIF-1-derived structures containing nucleocapsids were 100% based on VP39 signals (Fig. 7D). The ratios based on DAPI signals were over 80% but lower than those of VP39 signals (Fig. 7D), presumably due to the signal weakness. We further examined nucleocapsid invasion into tissues and found that numerous viral nucleocapsids were observed within the nucleus of the tracheal epithelial cell; this cell was adjacent to the virus-

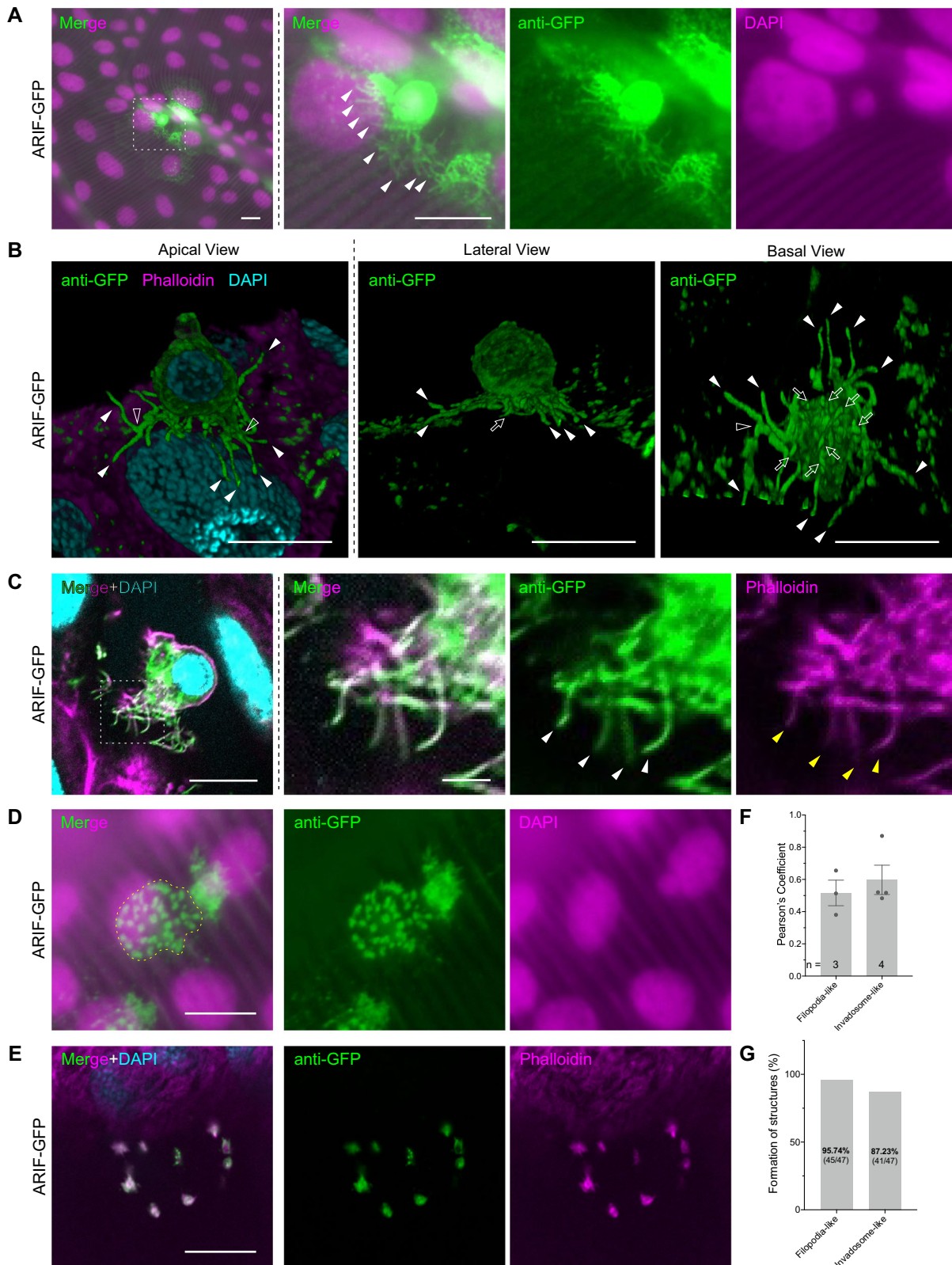

infected hemocyte with ARIF-1-derived structures and did not exhibit nuclear VP39 accumulation (this is necessary for progeny nucleocapsid production) (Fig. 7C, Supplementary Fig. S6). Thus, these nucleocapsids were likely produced within the attached hemocyte with nuclear VP39 accumulation, infected the adjacent cell, and moved into its nucleus without disassembly–this type of nuclear entry is same as AcMNPV[32,33]. 75.00% and

53.13% of the attached hemocytes showed nucleocapsid signals (VP39 and DAPI, respectively) in the adjacent tracheal epithelial cells (Fig. 7E), suggesting the high efficiency of tissue invasion in the invasive route.

Furthermore, we generated another recombinant BmNPV expressing mCherry-fused GP64 and examined its subcellular localization. GP64 is the membrane fusion protein of group I nucleopolyhedroviruses, essential for

**Fig. 4 | BmNPV-infected hemocytes form two types of ARIF-1-derived cell structures on the tracheal surface.** Immunostained *B. mori* larval trachea and hemocytes infected with ARIF-GFP were observed under zoom microscopy (**A, D**) or confocal microscopy (**B, C, E**) at 20–24 hpi. Anti-GFP (green), GFP-fused ARIF-1; DAPI (magenta in (**A, D**) and cyan in (**B, C, E**)), dsDNA; phalloidin (magenta in (**B, C, E**)), F-actin. Bar, 10 μm. **A** ARIF-GFP-infected hemocytes on the tracheal surface. A white dashed square is the area enlarged in the right panels. White arrowheads, filopodia-like protrusions. **B** 3D reconstruction of confocal microscopy images. Filled arrowheads, filopodia-like protrusions at the lateral sides; outline arrowheads, branching points of filopodia-like protrusions; outline arrows, invadosome-like protrusions at the basal side. **C** Localization of ARIF-1 and F-actin in filopodia-like protrusions (white and yellow arrowheads, respectively). Single-plane confocal images are shown. **D** Remnants of invadosome-like structures with aggregating ARIF-1. Yellow dashed line, a putative area of hemocyte attachment. **E** Localization of ARIF-1 and F-actin in invadosome-like structures. Single-plane confocal images are shown. **F** Pearson's colocalization coefficient of ARIF-1 and F-Actin in ARIF-1-derived structures. Data shown are mean ± SEM. **G** Percentage of attached hemocytes with ARIF-1-derived structures.

**Fig. 5 | SEM observation of attached hemocytes.** Mock- (**A**), hspGFP- (**B**), and hspGFP-ARIFD- (**C**) infected larval trachea and attaching hemocytes at 24 hpi were observed. The dashed line area in (**B**) is enlarged in (**B'**). Arrowheads, thin mesh-like structures; outline arrows, small vesicles. Bar, 5 μm (**A–C**) and 2 μm (**B'**).

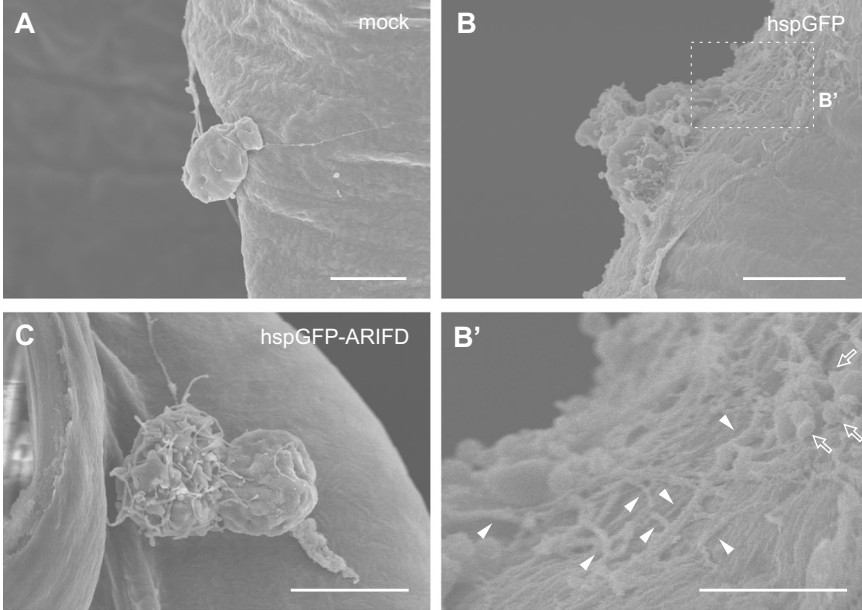

virus entry and cell-to-cell transmission[34,35]. We observed that mCherry-fused GP64 colocalized with ARIF-1 at both filopodia-like protrusions and invadosome-like structures on the cell surface (Supplementary Fig. S7). Taken together with our findings in this section, this result implies that viral nucleocapsids may bud as BVs from ARIF-1-derived structures and then enter into the adjacent tracheal epithelial cells via GP64-mediated endocytosis.

### Both the extracellular loop and C-terminal intracellular region are essential for the function of ARIF-1

Preceding studies reported that the C-terminal intracellular region of ARIF-1 plays a critical role in enhancing systemic infection in lepidopteran larvae[23] and inducing invadosomes in lepidopteran cultured cells[20]. To investigate the specific regions responsible for these functions, we generated recombinant BmNPVs expressing ARIF-1 with partial deletions in either the extracellular loop region (ARIFΔEXL-GFP) or the C-terminal intracellular region (ARIFΔC1-GFP, ARIFΔC2-GFP, and ARIFΔC3-GFP) (Fig. 8A). The protein expression and plasma membrane localization of these partially deleted ARIF-1 variants were verified in *B. mori* cultured cells (Supplementary Figs. S8, S9). Similar to the observation of hspGFP-ARIFD-infected samples (Fig. 3C), the number of attached hemocytes was very few upon infection with these ARIF-1 mutant BmNPVs, but we were able to observe a few attached hemocytes under confocal microscopy (Fig. 8B, upper panels). Only a few filopodia-like protrusions at most were formed in attached hemocytes infected with ARIFΔEXL-GFP, ARIFΔC1-GFP, and ARIFΔC2-GFP. In contrast, some ARIFΔC3-GFP-infected hemocytes formed many cell protrusions to attach to the tracheal surface, but their thin and straight shape appeared different from ARIF-1-derived filopodia-like protrusions. In addition, 3D reconstruction images revealed that none of these ARIF-1 mutants formed invadosome-like structures at the basal side (Fig. 8B, lower panels). These results suggest that the extra loop and the C-terminal intracellular regions are crucial for inducing ARIF-1-derived structures.

Next, we examined the viral propagation of ARIF-1 partial deletion mutants. In cultured cells, no significant difference was observed in viral DNA amount in infected cells (Supplementary Fig. S10A). BV production decreased 63–70% with a significant difference in ARIFΔC3-GFP compared to wild-type viruses (Supplementary Fig. S10B), implying that BV budding rather than viral DNA replication might be affected by partial deletion of ARIF-1 in this virus. In the case of larval infection, all the ARIF-GFP mutants showed more than a 30-fold lower number of OBs in the hemolymph compared to ARIF-GFP, while they did not show a significant difference when compared to the ARIF-1-deletion BmNPV (ΔMetARIF) (Fig. 8C, D). These findings suggest that these mutant ARIF-1 proteins lack the ability to enhance systemic infection. Collectively, the ARIF-1's ability to induce filopodia-like protrusions and invadosome-like structures is presumably crucial for enhancing systemic infection.

In summary, our results suggest that BmNPV remodels host hemocytes to form ARIF-1-derived structures for attaching to the tissue surface and then breach the BL barrier to deliver progeny virions directly into the tissues. This ARIF-1-mediated hemocyte remodeling greatly contributes to the quick spread of virus infection throughout the host body.

### Discussion

Many pathogenic or parasitic organisms are well-recognized for manipulating host cytoskeletal systems for their benefit[2,3,5,36,37]. ARIF-1 has been reported as an inducer of actin rearrangement in lepidopteran cultured cells after alphabaculovirus infection[18]. Recently, ARIF-1's invadosome-inducing activity was demonstrated[20], but its biological significance in the viral life cycle remains largely unknown. In the present study, we found that ARIF-1 is crucial for the attachment of BmNPV-infected hemocytes to the tissue

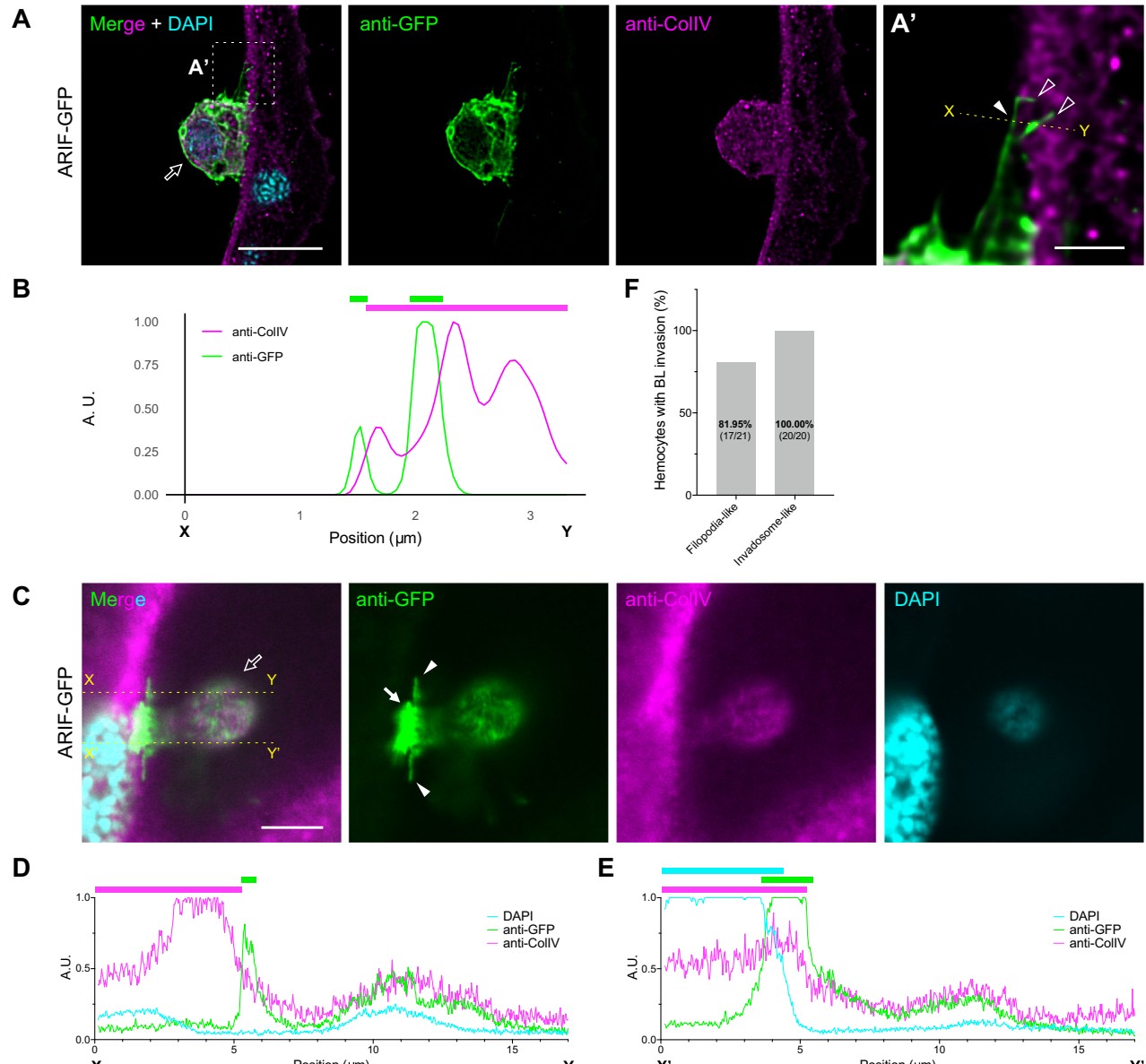

**Fig. 6 | ARIF-1-derived structures invade the BL barrier.** Immunostained ARIF-GFP-infected hemocytes on the trachea at 24 hpi were observed under confocal microscopy. Single-plane confocal images of the invasion of filopodia-like protrusions (**A**) and invadosome-like structures (**C**) into the BL barrier are shown. A white dashed square in (**A**), the area enlarged in (**A'**); white outline arrows, attached hemocytes; white filled and outline arrowheads, filopodia-like protrusions on the tracheal surface and those invading the BL, respectively; white filled arrows, invadosome-like structures. Anti-GFP, GFP-fused ARIF-1 (green); anti-ColIV,

collagen IV (a BL component, magenta); DAPI, dsDNA (cyan). Bar, 10 µm in (**A, C**) and 2 µm in (**A'**). LIGHTNING deconvolution was applied in (**A**).
**B, D, E** Fluorescence intensity plot. Relative fluorescent intensity of yellow dotted lines X–Y in (**A'**) and X–Y and X'–Y' in (**C**) are shown in (**B**), (**D**), and (**E**), respectively. The top bars indicate the range of ARIF-1-derived structures (green), tracheal epithelia (magenta), and the nucleus (cyan). (**F**) Percentage of attached hemocytes exhibiting BL invasion by ARIF-1-derived structures. The numbers of hemocytes observed are shown in parentheses.

surface and the remodeling of the host cytoskeletal system to form two types of ARIF-1-mediated structures (filopodia-like protrusions and invadosome-like structures). Based on our findings, we proposed the working model of the invasive route of BmNPV infection (Fig. 9): After infection, circulating hemocytes express ARIF-1 to attach to the tissue surface, forming filopodia-like protrusions and invadosome-like protrusions. Filopodia-like protrusions extend to the lateral side along with the BL surface, sometimes branch off, and penetrate the BL barrier. They may also transform into widely spreading mesh-like structures. Invadosome-like structures are dot-like aggregations at the basal side and their extension into the BL. Both structures can penetrate the BL barrier and deliver nucleocapsids into the BL-protected tissues, resulting in the efficient spread of infection. While our microscopic observations clearly demonstrate this

working model, this does not fully explain the large size of the infection foci formation as early as 24 hpi because 24 hpi is, in general, the timing in that baculoviruses have just completed one replication cycle and begun to infect surrounding cells. We speculated that ARIF-1-derived structures may not only penetrate the BL barrier but also degrade it in the neighboring area. Invadosomes in mammals are known to secrete proteinases to degrade the ECM[38], implying that ARIF-1-derived structures may also secrete such enzymes to degrade distant BL. In vitro analysis showed abundant expression of ARIF-1 from 8 hpi (Supplementary Fig. S2B). Considering our SEM observation of thin mesh-like structures spreading in a wide area from attached hemocytes (Fig. 5), these structures possibly breach the broad area of the neighboring BL barrier, and hence circulating BVs (remaining of injection) can establish tissue infection at very early timing after

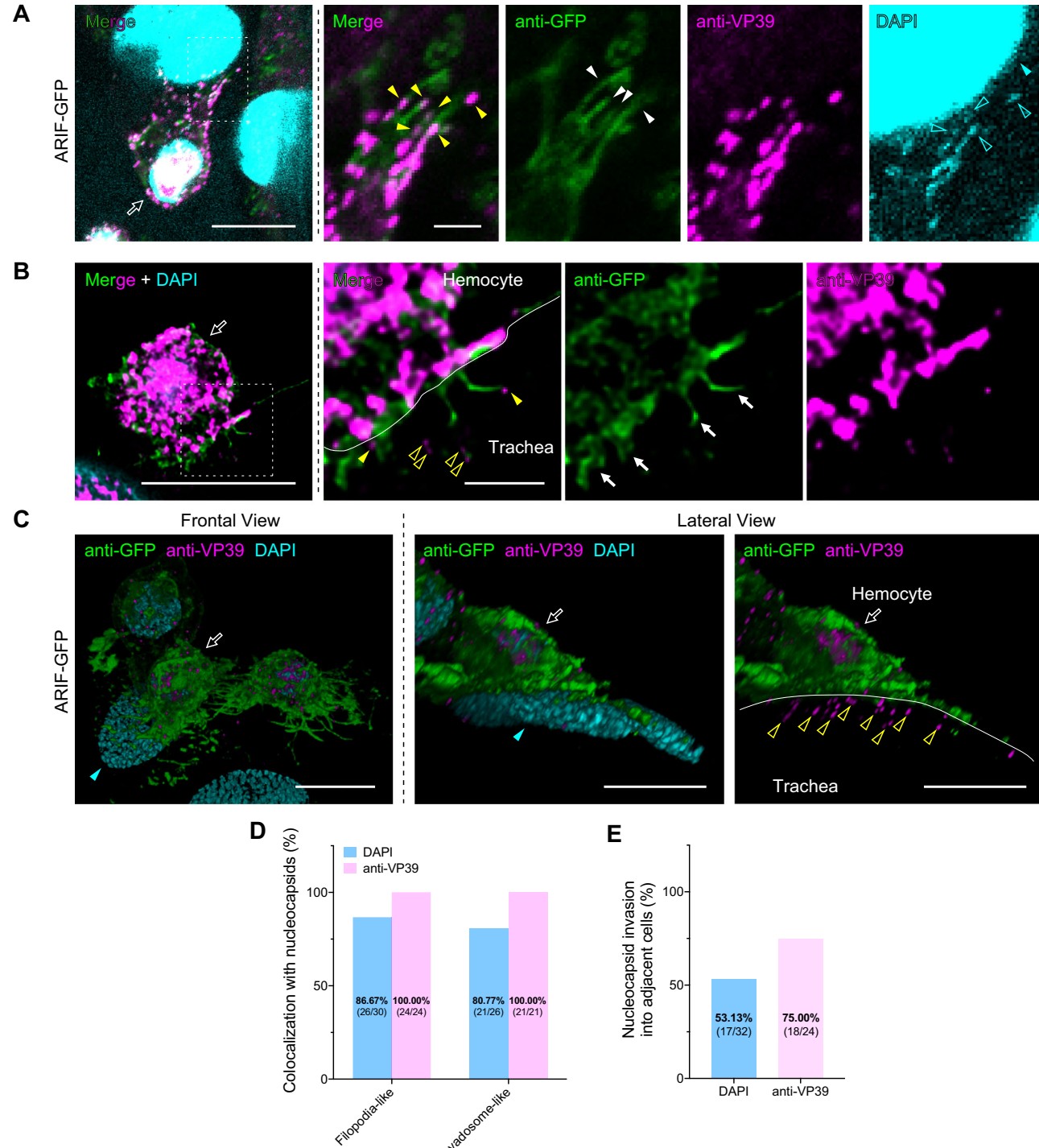

**Fig. 7 | ARIF-1-derived structures work as migratory paths for viral nucleo-capsids.** Immunostained ARIF-GFP-infected hemocytes on the trachea at 24 hpi were observed under confocal microscopy. **A**, **B** Nucleocapsids in filopodia-like protrusions (**A**) and invadosome-like structures (**B**) (single plane). White dashed rectangles, the regions enlarged in right panels. **C** 3D reconstruction images of nucleocapsids invading the adjacent cell. In (**A–C**), white outline arrows, attached hemocytes; white filled arrowheads and arrows, filopodia-like protrusions and invadosome-like structures, respectively; yellow filled and outline arrowheads, nucleocapsids in ARIF-1-derived structures and those invading in adjacent tracheal cells, respectively; cyan filled arrowheads, the nuclei of tracheal epithelial cells; cyan outline arrowheads, viral dsRNA in nucleocapsids; white lines, borders of the trachea and hemocytes. Anti-GFP, GFP-fused ARIF-1 (green); anti-VP39, VP39 (a major capsid protein, magenta); DAPI, dsDNA (cyan). Bar, 10 μm in (**C**) and left panels of (**A**, **B**), and 2 μm in right panels of (**A**, **B**). LIGHTNING deconvolution was applied in (**B**, **C**). **D** Percentage of attached hemocytes that have ARIF-1-derived structures with nucleocapsid colocalization. **E** Percentage of attached hemocytes whose adjacent tracheal epithelial cells have nucleocapsid signals. In (**D**, **E**), the number of hemocytes observed is shown in parenthesis. DAPI and anti-VP39 signals were used as nucleocapsid markers.

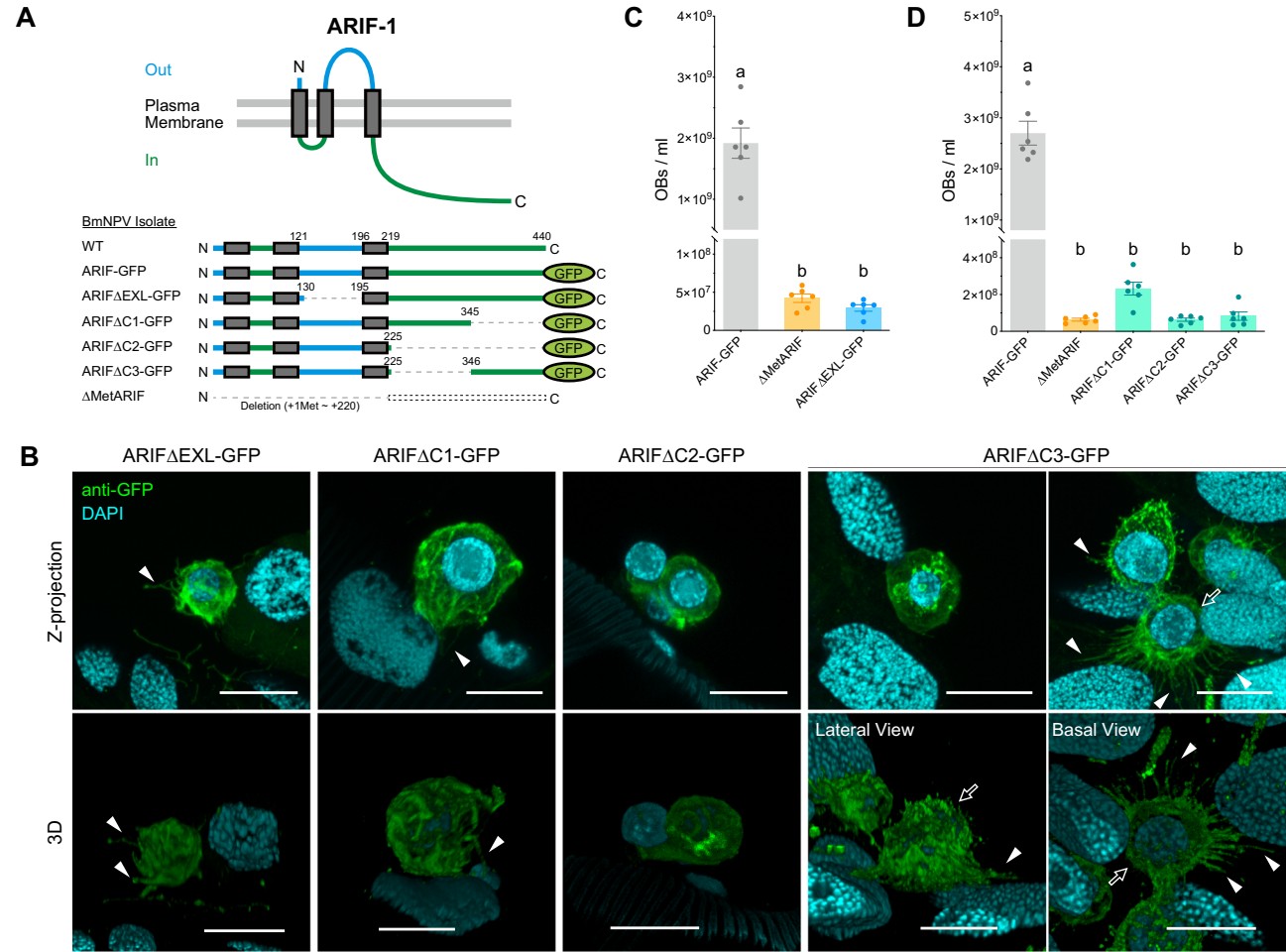

**Fig. 8 | Deletion of the extracellular loop and C-terminal region impairs the function of ARIF-1. A** Schematic images of ARIF-1, GFP-fused ARIF-1 (ARIF-GFP), and truncated variants of ARIF-GFP expressed by recombinant BmNPVs. **B** Z-projection (upper panels) and 3D reconstruction (lower panels) of confocal images from virus-infected hemocytes expressing truncated ARIF-GFP variants on the tracheal surface. Anti-GFP, GFP-fused ARIF-1 variants (green); DAPI, dsDNA (cyan). Bar, 10 μm. White arrowheads, filopodia-like protrusions. White outline arrows indicate the same hemocyte. **C, D** The number of OBs released in the hemolymph of virus-infected larvae at 96 hpi. Data shown are mean ± SEM ($n = 6$). Different letters indicate statistically different groups ($p < 0.05$, one-way ANOVA with Tukey's multiple comparisons test).

intrahemocoelic infection. Future time-course observation and investigation of the BL degradation will disclose the mechanism of the quick infection foci formation via ARIF-1.

Invadosomes (also known as podosomes or invadopodia) are 0.5–2 μm in diameter dot-like actin structures[39–41] that may be organized into dynamic ring- or rosette-like clusters[42,43]. Invadosomes have been shown to form in a range of cell types, including mammalian monocyte-derived cells[39], smooth muscle cells[44], and endothelial cells[45]. Vertebrate tumor cell lines can also form invadosomes, allowing them to modify the ECM and undergo metastasis[46,47]. Human immunodeficiency virus 1 (HIV-1) is known to increase the stability and proteolytic activity of host macrophage invadosomes to enhance their mesenchymal migration[4]. In contrast to mammalian cells, the biological role of invadosomes in invertebrates has been largely unknown, except for the BL transmigration of anchor cells in the nematode *Caenorhabditis elegans* that establish the initial connection between the uterine and vulval epithelium[48]. Our study, to the best of our knowledge, first demonstrated the biological importance of virus-induced invadosomes in vivo in invertebrate species. This implies that invadosome hijacking is a highly effective strategy for systemic infection and thus convergently evolved among pathogenic agents of diverse animal species. Insect hemocytes are primarily responsible for cellular immunity (e.g., nodulation, encapsulation, and phagocytosis) against pathogenic invasion[49,50], but a recent study found that immune responses in glial cells trigger macrophage

infiltration into the central nervous system (CNS) of the fruit fly *Drosophila melanogaster*[51]. This implies that insect hemocytes also have the ability to cross the ECM barrier. Because ARIF-1-induced invadosomes show direct ECM degradation ability in in vitro cultured cells[20] and indeed penetrate the BL barrier (Fig. 6C), invadosome-like structures of alphabaculovirus-infected hemocytes presumably recruit or activate matrix metalloproteases (MMPs) or other types of proteases to degrade ECM components like vertebrate invadosomes[52,53]. Further research into the molecular properties of insect invadosome components will offer insights into the nature of invadosomes in invertebrate species and how viruses manipulate them.

Some pathogenic proteins for cytoskeletal rearrangement share similarities with host cytoskeleton-related proteins[37]. For example, the baculovirus P78/83 protein induces actin polymerization through its WASP-like domain to drive motility for intracellular movement of viral nucleocapsids[33]. ARIF-1 is a unique type of protein for cytoskeletal manipulation since it lacks an evidently homologous region with such host or pathogenic proteins. ARIF-1 possesses a proline-rich area in the C-terminal intracellular region; proline accounts for 25.7% of the C-terminal amino acid region 277–430 of BmNPV ARIF-1. In *Spodoptera frugiperda* Sf21 cells, aa 303–371 of AcMNPV ARIF-1–this corresponds to aa 306–374 of BmNPV ARIF-1–is sufficient to induce invadosomes[20]. Actin-related proteins containing SH3, WW, and EVH1 domains are known to bind to proline-rich areas[54–58]. Furthermore, ARIF-1 is tyrosine phosphorylated[24], which is

**Fig. 9 | Schematic image of alphabaculovirus invasion into the host tissues from the hemolymph**.

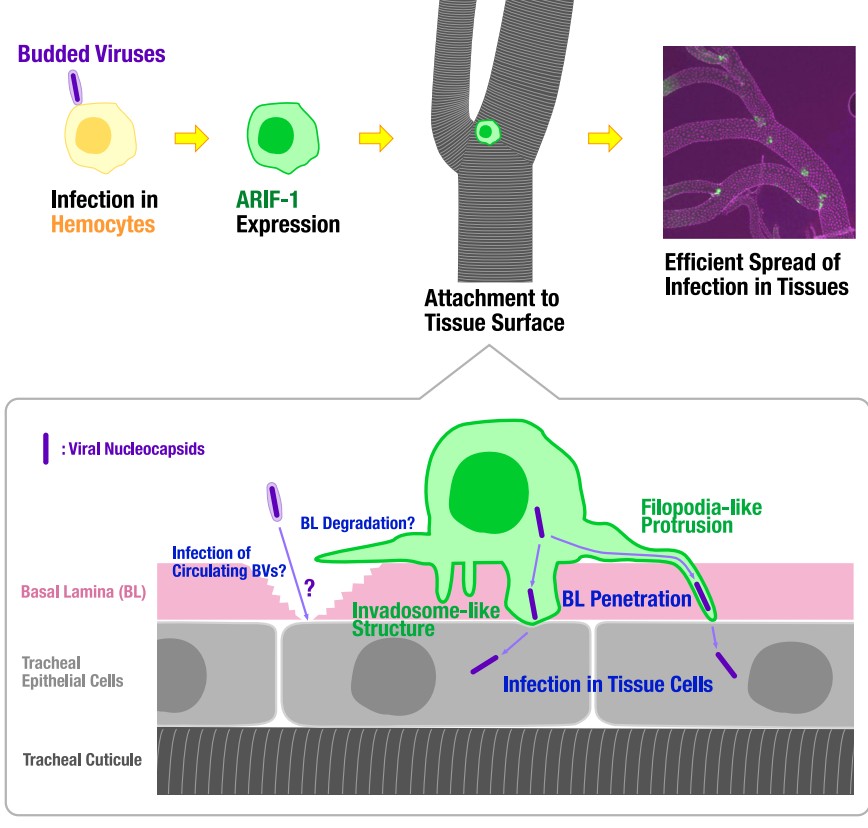

presumably crucial for its function because an amino acid substitution at the putative phosphorylation site (Y332F) of AcMNPV ARIF-1 causes loss of invadosome-inducing ability[20]. Phosphorylated tyrosine residues are important components of SH2 domain binding sites[59,60], widely found in proteins that regulate the actin cytoskeleton. ARIF-1, as a whole, may induce invadosome-like structures by interacting with actin-related proteins via its C-terminal proline-rich region and phosphorylated tyrosine residue(s). In contrast to the invadosomes, no previous studies have shown the link between ARIF-1 and filopodia-like protrusions. Filopodia-like protrusions appear to be ARIF-1-derived structures rather than cell attachment byproducts, because partial deletion in the C-terminal region of ARIF-1 caused markedly fewer filopodia-like protrusions in virus-infected hemocytes, even on the tracheal surface (Fig. 8B). Considering their BL-invading ability (Fig. 6A), filopodia-like protrusions have different characteristics from typical filopodia in healthy host cells and those previously reported to be induced by other viruses[3]. Taken together, ARIF-1 induces two types of actin cytoskeleton rearrangement, filopodia-like protrusions and invasion into tissues via invadosome-like structures, whose commonalities and differences are presently unknown. Furthermore, we found that ARIFΔEXL-GFP, which expresses ARIF-1 lacking the extracellular loop region, also induced markedly fewer filopodia-like protrusions (Fig. 8B). The ARIFΔEXL-GFP protein was colocalized with F-actin at the plasma membrane (Supplementary Fig. S9). These results imply that associating with extracellular factors (e.g., ECM component proteins) is necessary for forming ARIF-1-derived structures. Further analysis of the membrane topology and ECM-interacting ability of this mutant ARIF-1 protein will clarify the molecular mechanism of ECM recognition by ARIF-1.

In the BmNPV–*B. mori* combination, hemocyte infection has been considered important for the systemic spread of viral infection. *B. mori* hemocytes are highly permissive to BmNPV infection. Previously, we reported that the ratio of BmNPV-infected hemocytes reached over 50% in both intrahemocoelical and oral infection; at this time, viral infection started to spread within host tissues[61]. In addition, the BmNPV genome encodes the viral fibroblast growth factor (*vfgf*)–a gene homologous to the mammalian *fgf*–that is secreted into the extracellular space and then induces the chemotaxis of host cells via the host's FGF receptor Breathless[62–64]. Combined with in vivo analyses using *fgf* knock-out viruses, the model of vFGF's molecular function was proposed; vFGF may facilitate systemic BmNPV infection by stimulating the migration of uninfected hemocytes to infection foci[61,65]. Considering our findings on the ARIF-1's ability for hemocyte actin remodeling, ARIF-1 and vFGF are likely to facilitate synergistically hemocyte-mediated systemic infection of BmNPV; after primary midgut infection, uninfected hemocytes are gathered by virus-infected cells via vFGF, and then the hemocytes are infected, circulated in the hemolymph, and finally attached to the tissue surface for breaching the BL barrier. *Arif-1* is conserved in most alphabaculoviruses (lepidopteran-specific nucleopolyhedroviruses)[18,19], raising the possibility that ARIF-1's function is conserved among alphabaculovirus species. Keddie et al. [17] pointed out that hemocyte infection seemed to be important for systemic infection in AcMNPV-infected *T. ni* larvae, based on the histological observation of virus-infected hemocytes on the epidermis at the early stage of infection[17]. We speculated that this photo might have captured the ARIF-1-mediated hemocyte attachment in AcMNPV-infected *T. ni* larvae. Meanwhile, the importance of hemocytes in the systemic infection presumably depends on the baculovirus–host combination. Larval hemocytes of *Helicoverpa zea*, a semi-permissive host of AcMNPV, are highly resistant to AcMNPV infection and participate in the host's melanization and encapsulation response to the virus-infected foci[66]. Similarly, hemocytes of *S. frugiperda* and *Spodoptera littoralis* are also resistant to AcMNPV infection; therefore, AcMNPV infection mainly spreads via the tracheoblast-mediated evasive route[67,68]. Given that *H. zea*, *S. frugiperda*, and *S. littoralis* belong to different subfamilies in the family Noctuidae from the subfamily Plusiinae–*T. ni* and *A. californica* (the AcMNPV's originally isolated host) belong to this subfamily–, AcMNPV may have evolved to evade the immune system of Plusiinae species, which presumably enables them to exploit hemocytes for systemic infection. Collectively, the present study clearly demonstrates the

previously uncertain molecular mechanism of hemocyte-mediated tissue invasion of alphabaculoviruses. The deletion of *arif-1* caused a significant delay in progeny virus production but did not lead to reduced OB levels in terminal larval carcasses (Supplementary Fig. S11). However, *arif-1* deletion led to failure in manipulating host behavior[23], which means that the success or failure of hemocyte-mediated infection may have a striking impact on viral dissemination in the environment. Furthermore, some arthropod-borne viruses are considered to utilize hemocytes to enter the salivary gland of host insects, although their molecular mechanisms remain unknown[7,69]. Our findings may provide clues for elucidating the mechanism of hemocyte-mediated tissue entry by the wide range of insect-related viruses.

In the past two decades, many studies have reported that some populations of insect hemocytes are sessile, i.e., weakly adhere to the surface of internal organs[26–30]. In *D. melanogaster* larvae, for example, embryo-derived plasmatocytes and crystal cells form sessile clusters in subepidermal locations called "sessile pockets," which serve as a source of additional hemocyte circulation against parasitoid wasp infestation[29,70]. In this study, we discovered hemocyte attachment to tissue surfaces in BmNPV-infected *B. mori* larvae. Considering that the number of hemocytes attaching to the trachea is remarkably low in the mock group (Fig. 3C), we speculate that these attached hemocytes are not derived from sessile hemocytes but rather from circulating hemocytes, which are newly attached as a result of ARIF-1 expression. Over 88% of the attached hemocytes were granulocytes, a higher percentage than the reported granulocyte proportion among circulating hemocytes in silkworm larvae (50–80%)[25]. Additionally, the formation of a mesh-like structure and the attachment of small vesicles in the SEM observation (Fig. 5) suggest some similarities to granulocyte degranulation. Considering that ARIF-1's effect on the actin cytoskeleton differs among cell types or culture conditions (earlier studies[18,20,23,71] and this study) and hence ARIF-1 appears to have cell type-specific roles, BmNPV might promote attachment to tissue surfaces by hijacking a granulocyte-specific immune mechanism using ARIF-1. On the other hand, it is also evident that ARIF-1-mediated hemocyte attachment is not necessarily a granulocyte-specific phenomenon because approximately 15% of the virus-infected attached hemocytes are Hemocytin-negative (Fig. 3E). BmNPV can infect all types of circulating hemocytes (granulocytes, spherulocytes, plasmatocytes, and oenocytoids), but plasmatocytes were more resistant to BmNPV infection[72]. Detailed observations of each hemocyte type will be essential to elucidate the relationship between ARIF-1's function and hemocyte types.

In this study, we found that ARIF-1 facilitates fat body infection by increasing the number and size of infection foci. The size of hspGFP infection foci exhibited two peaks in the fat body (Fig. 2D), and the ratio of the larger infection foci significantly increased than that of hspGFP-ARIFD (Fig. 2E). Therefore, ARIF-1 may enable BmNPV to establish the invasive infection in the fat body, in addition to the tracheoblast-mediated evasive infection with smaller infection foci. Based on the findings on the hemocyte-mediated invasive infection in the central tracheal region, we consider two possible mechanisms for facilitating fat body infection: (i) virus-infected hemocytes are attached to the tracheoblasts or tiny tracheal branches on the fat body, and then progeny viruses are delivered into the BL-surrounded space of tracheal epithelia and adjacent fat body. (ii) virus-infected hemocytes directly attach to and breach the fat body BL. It is difficult to distinguish these two infection routes because the fat body is wired with numerous tiny tracheal branches. Similarly, other tissues with the BL barrier may also be the targets of hemocyte attachment, but at present, we do not have convincing data about tissue preference. Further histological analysis of various tissues with cell type-specific antibodies–the repertoire of such available antibodies is not enough at present in lepidopteran insects–will provide insights into this question.

Currently, one plausible theory of the midgut escape is that alphabaculoviruses overcome the midgut BL barrier by escaping to the hemocoel through tracheoblast infection[15], as described in the "Introduction" section and Supplementary Fig. S1. In addition, vFGF seems to facilitate the tracheoblast-mediated midgut escape route in AcMNPV-infected *T. ni* larvae by attracting tracheoblast migration towards infection foci and

inducing BL degradation[9,73]. While this tracheoblast-mediated route seems to be an important midgut escape route in alphabaculoviruses, an observation made using electron microscopy revealed AcMNPV nucleocapsids pushing the BL from the basal side of midgut epithelial cells and those released on the hemocoelic side at the initial stage of oral infection, suggesting that alphabaculoviruses may also directly cross the midgut BL[13]. Since AcMNPV highly expresses *arif-1* during the initial stage of midgut infection[74], ARIF-1-mediated cytoskeletal modification may also occur in the midgut. *arif-1* is conserved in all Group I and most Group II alphabaculoviruses (systemic infection), whereas this gene is absent in betabaculoviruses (systemic, midgut and fat body-specific, or midgut-specific infection) or gamma- and deltabaculoviruses (midgut-specific infection)[14,18,19]. If ARIF-1 causes invadosome development on the basal side of the midgut epithelial cells, this might be the alternative method of direct midgut crossing in alphabaculoviruses that have *arif-1*. Many other insect-borne pathogens have midgut escape pathways, some of which use the cytoskeletal system in host midgut epithelial cells[1,2,7]. However, we could not locate any reports of such pathogens inducing invadosomes. Our research offers a novel strategy for the midgut escape in invertebrate hosts.

## Methods

### Insects, cells, and viruses

*B. mori* larvae (F1 hybrid Kinshu × Showa) were reared following previously described methods[23]. BmN (BmN-4) cells were cultured at 26 °C in TC-100 medium (AppliChem) supplemented with 10% fetal bovine serum[75]. The T3 strain[76] of BmNPV was used as the wild-type virus (WT). ARIFDR (BmARIFDR) represents a revertant virus of the *arif-1* deletion mutant BmNPV, ARIFD (BmARIFD)[23]. ARIF-FS (BmARIF-FS) is an *arif-1*-deficient mutant containing a frame-shift mutation at the amino acid residue 346 of ARIF-1[23]. hspGFP (BmhspGFP) is a recombinant BmNPV expressing GFP under the control of the *Drosophila hsp70* promoter, which can induce gene expression from early to very late stages of BmNPV infection[72]. hspGFP-ARIFD (BmhspGFP-ARIFD)[23] also contains the *hsp70-gfp* cassette at the *polyhedrin* (*polh*) locus, with *arif-1* disrupted by inserting the *hsp70-lacZ* (β-galactosidase) cassette[77]. GFP expression levels of hspGFP and hspGFP-ARIFD are sufficient to distinguish virus-infected cells throughout the infection stages, although hspGFP-ARIFD showed 30% reduced GFP fluorescence than hspGFP in cultured cells (Supplementary Fig. S12); this might be because hspGFP-ARIFD has two *hsp70* promoters in the *hsp70-gfp* and *hsp70-lacZ* cassettes. Virus titers (plaque-forming units; PFU) of WT and mutant BmNPVs were determined by plaque assay on BmN cells[78].

### Virus infection in *B. mori* larvae

Fifth- or fourth-instar day-0 *B. mori* larvae were starved for several hours and injected with 50 μL (fifth-instar) or 25 μL (fourth-instar) viral suspension containing BVs ($1 \times 10^6$ PFU for immunostaining; $5 \times 10^5$ PFU for SEM; $1 \times 10^5$ PFU for other experiments) and 5 mg/mL kanamycin using a syringe fitted with a 30-gauge needle (Nipro, Japan). The larvae were inoculated within 24 h after molting. Then, the larvae were returned to the artificial diet and reared at 25 °C[79].

### Immunostaining of virus-infected tissues

For immunostaining of the trachea and fat body, tissues were dissected from virus-infected larvae at specific time points in phosphate-buffered saline (PBS; 137 mM NaCl, 8.10 mM $Na_2HPO_4$, 1.47 mM $KH_2PO_4$, 2.68 mM KCl, pH 7.4). When we dissected samples for quantitative data (e.g., the number of attached hemocytes, the ratio of ARIF-1-derived structure formation, etc.), we added benzamidine hydrochloride (final conc. 10 mM; Sigma-Aldrich, #B6506) to ice-cold PBS to inhibit serine protease-mediated immune reactions[80,81]. Each tracheal sample consisted of tracheal tubules from a single spiracle. The tissue samples were fixed in 3.7% paraformaldehyde (PFA) for RT, 30 min or 4 °C, O/N. Throughout each incubation step, the samples in 1.5 mL tubes were gently tapped to ensure proper mixing and avoid detachment of hemocytes. Subsequently, the tissues were washed three times in 0.1% bovine serum albumin (BSA)/PBS,

permeabilized in 0.5% Triton X-100 in 0.1% BSA/PBS for 30 min, washed three times in 0.1% BSA/PBS, and blocked in 1% BSA/PBS for RT, 1 h or 4 °C, >O/N until all the time course samples were ready to be taken through to the subsequent steps simultaneously. Then, the samples were incubated overnight at 4 °C with primary antibodies (see below). After three time washes in 0.1% BSA/PBS, the cells were incubated overnight at 4 °C with secondary antibodies (see below). Following three washes in 0.1% BSA/PBS, the cells were stained with -Cellstain- DAPI solution (1:500; DOJINDO, Japan) and optionally with Rhodamine phalloidin (1:20; Invitrogen) in 0.1% BSA/PBS for 30 min. Subsequently, the cells were washed three times in 0.1% BSA/PBS and sealed in ProLong Gold Antifade Reagent (Invitrogen), ProLong Diamond Antifade Mountant (Invitrogen), or SlowFade Glass Antifade Mountant (Invitrogen). The primary antibodies used were rabbit anti-GFP antibody (1:400; MBL, Japan, #598), mouse anti-GFP(15) monoclonal antibody (1:100–1:400; Santa Cruz, #sc-101525), and chicken anti-GFP antibody (1:200–1:500; abcam, # ab13970) for GFP, rabbit anti-Hemocytin antiserum (1:600–1:1000; provided by Dr. Ryoichi Sato)[82] for Hemocytin, rabbit anti-Collagen IV (ColIV) antibody (1:50–1:200; abcam, #ab6586) for ColIV, rabbit anti-DsRed antibody (1:200; Clontech, #632496) for mCherry-fused GP64, and mouse anti-VP39 antibody (1:100–1:400; provided by Dr. Loy E. Volkman)[83] for VP39. The same antibody dilution was used in the samples of each experiment. The secondary antibodies used were Alexa Fluor 488-conjugated goat anti-rabbit antibody (1:400; Invitrogen, #11070), Alexa Fluor 488-conjugated goat anti-mouse antibody (1:400; Invitrogen, #A11001), Alexa Fluor 488-conjugated goat anti-chicken antibody (1:200; Invitrogen, #A-11039), Alexa Fluor 546-conjugated goat anti-rabbit antibody (1:400; Invitrogen, #11071), or Alexa Fluor 546-conjugated goat anti-mouse antibody (1:400; Invitrogen, #11003). In the antibody reaction steps, antibodies were diluted in 0.1% BSA/PBS except for the anti-ColIV staining of Fig. 6A, in which Can Get Signal Immunostain Solution B (TOYOBO, Japan) was used for amplifying signals. To observe remnants of invadosome-like structures, hemocytes attached to the tracheal surface were disrupted by gently crushing the samples with cover glasses. Immunostained samples were examined using the on-axis zoom microscope Axio Zoom.V16 with objective lens PlanNeoFluar Z 2.3x/0.57 FWD 10.6 mm (ZEISS) and ZEN software ver. 2.3 (ZEISS), confocal scanning laser microscopes Nikon C1Si with EZ-C1 software Gold Version 3.60 build 770 (Nikon, Japan; objectives were Plan Apo VC 100x/1.40 Oil or PlanFluor ELWD 20x/0.45, and lasers were 405 nm for DAPI, 488 nm for Alexa Fluor 488, and 561 nm for Rhodamine, and filters were 450/35 nm for DAPI, 515/30 nm for Alexa Fluor 488, and 650 nm long pass for Rhodamine) (Figs. 4C, 7A, Supplementary Figs. S2D, S9), FLUOVIEW FV10i with FV10i software ver.2.1.1.7 (Olympus, Japan) (Fig. 6C, Supplementary Figs. S5, S7), or Leica STELLARIS 5 WLL (Leica) with 405 nm laser for DAPI and Leica Application Suite X Version 4.4.0.24861 (Leica) (Figs. 3A, B, 4B, E, 6A, 7B, C, 8B, Supplementary Figs. S4B, S6). The LIGHTNING module in the STELLARIS 5 platform was used for the deconvolution of confocal images (Figs. 6A, 7B, C, Supplementary Figs. S4B, S6).

For the immunostaining of isolated hemocytes, a PAP pen was used to draw 1 × 1 cm squares on Poly-L-Lysine coated slide glasses. At 18 hpi, hemolymph from mock- or virus-infected larvae were collected into 1.5 mL tubes with a small amount of phenylthiourea to avoid melanization, mixed by pipetting, mounted on the square portions of slide glasses, and let stand for 10 min to allow hemocytes to stick to the glass surface. The hemolymph supernatant was carefully removed using a micropipette. The hemocytes were then fixed in 3.7% PFA/PBS for 10 min, washed three times in TBS (25 mM Tris, 137 mM NaCl, 2.7 mM KCl, pH 7.4), and permeabilized for 5 min in 0.1% Triton X-100 in TBS. After three TBS washes, samples were blocked for 15 min in 1% BSA/PBS, rinsed in TBS, and incubated O/N at 4 °C with the primary antibody (rabbit polyclonal anti-phospho Histone H3 (Ser10) antibody (anti-pH3), 1:200; Sigma-Aldrich, #06-570). Following three TBS washes, the cells were incubated for 1 h with the secondary antibody (Alexa Fluor 488-conjugated goat anti-rabbit antibody, 1:400; Invitrogen, #11070). After three TBS washes, the cells were stained for 15 min in TBS with -Cellstain- DAPI solution (1:400), rinsed three times in TBS, and then sealed in 10 mL of ProLong Gold Antifade Reagent, and examined using the FLoid Cell Imaging Station (Thermo Fisher Scientific).

Unless otherwise specified, all the incubations were carried out at room temperature.

## Image analysis

Fiji software[84] and Adobe Photoshop CS or 2025 (Adobe Systems Inc.) were used to process the images. For graph creation and statistical analysis, we used the Prism 8 software (GraphPad) or R (ver. 4.4.0)[85] with the ggplot2 package (version 3.5.1)[86]. All the fluorescent images were brightness-adjusted with no gamma correction, background signal cutoff, or alteration of signal ranges unless otherwise described. Such adjustment was applied equally across the entire image and to controls. The fluorescent images were pseudocolored with the colors shown in the figures or figure legends.

To visualize the spread of infection, zoom microscopic pictures of immunostained hspGFP- or hspGFP-ARIFD-infected trachea were manually stitched to display the whole region of each tracheal sample. The anti-GFP-immunostained pictures (images of virus-infected cells) were binarized and pseudocolored in magenta using the same threshold in all the photos to eliminate background signals. Signals from attached tissue debris and background signals from the tracheal cuticle (particularly the cut ends of tracheal tubules) were manually eliminated. Level-adjusted DAPI-stained pictures were handled using outline extraction. The photo's colors were then converted to grayscale, inverted, and pseudocolored in magenta. To show the shape of tracheal samples, these DAPI-derived pictures (gray) were placed underneath images of virus-infected cells (magenta).

The number and area of infection foci were determined from binarized anti-GFP-immunostained pictures of immunostained hspGFP- or hspGFP-ARIFD-infected tissues using the particle analyzer in Fiji. Only the infection foci with an area of more than 100 μm$^2$ were used for the analysis to remove noises. We used the following approach to compute the areas of tracheal samples: The tissue outlines from bright-field pictures were extracted and binarized. The samples' silhouettes were then manually made using Adobe Photoshop to calculate the areas using a particle analyzer in Fiji. Finally, the number of infection foci per area was calculated. Each tissue sample was obtained from a different individual larva. The number of infection foci over and under 1000 μm$^3$ was analyzed by chi-square test. We analyzed 4–5 pictures from a single tissue per virus.

The number of attached hemocytes to tracheal samples was counted under zoom microscopy Axio Zoom.V16. We counted either GFP- or Hemocytin-positive hemocytes and those with both signals. We did not count hemocytes that had neither GFP nor Hemocytin signals because it is challenging to find them by DAPI signals. We excluded hemocytes attached near the cutting sites and aggregated by Hemocytin from counting because they are presumably artifacts by dissection. Bright-field images of the tracheal samples were taken to process them in Adobe Photoshop 2025 and to calculate the samples' area in Fiji. Finally, the number of attached hemocytes per area was calculated. Each tissue sample was obtained from a different larva (n = 5).

To calculate the number of hemocytes in the mitotic stage (phosphorylated Histone H3 (pH3)-positive), fluorescent microscopic images of hemocytes were binarized with appropriate thresholds and treated with the Watershed algorithm to separate cells in close contact. The number of pH3-positive cells was counted using the particle analyzer in Fiji. Similarly, the total cell number was counted using DAPI fluorescence images. The ratio of mitotic hemocytes was derived from these data. Nine images each from two hemolymph samples were processed.

Z-projection, orthogonal view, and 3D reconstruction images of confocal microscopy were generated using Leica Application Suite X Version 4.4.0.24861 (Leica).

Coloc 2 Plugin ver.3.1.0 (https://imagej.net/plugins/coloc-2) was used for the colocalization analysis in Fiji. The area of filopodia-like protrusions and invadosome-like structures were manually selected in single optical planes of ARIF-GFP and phalloidin, and Pearson's colocalization coefficient was calculated. The mean value of the Pearson's coefficients of areas from an

individual hemocyte was regarded as the Pearson's coefficient of the hemocyte.

The ratio of the formation of ARIF-1-derived structures, their BL invasion, colocalization with nucleocapsids, and nucleocapsid invasion into adjacent cells were calculated by investigating Z-stack confocal images of attached hemocytes. The number of hemocytes observed was as follows; ARIF-1-derived structures, 47 hemocytes from 6 larvae; BL invasion, 21 (filopodia-like) and 20 (invadosome-like) hemocytes from 3 larvae; colocalization with nucleocapsids, 30 (filopodia-like, DAPI) and 26 (invadosome-like, DAPI) hemocytes from 4 larvae and 24 (filopodia-like, VP39) and 21 (invadosome-like, VP39) hemocytes from 3 larvae; nucleocapsid invasion into adjacent cells, 32 (DAPI) and 24 (VP39) hemocytes from 4 and 3 larvae, respectively.

The fluorescent intensity of confocal images was measured using Fiji.

### Larval bioassays

To measure the concentration of hemocytes, the hemolymph of mock- or virus-infected larvae was collected at 24 hpi into 1.5 mL tubes with a small amount of phenylthiourea to prevent melanization. The number of hemocytes was counted using a hemocytometer to calculate the concentration.

Body weights of mock- or virus-infected larvae were measured by an electronic scale at 24 hpi.

### Generation of recombinant BmNPVs

All the ARIF-1 recombinant BmNPVs were generated by homologous recombination of recombinant donor vectors and the genomic DNA of ARIFD. In ARIFD, the *arif-1* gene is disrupted by inserting the *hsp70-lacZ* (β-galactosidase) cassette that contains a Bsu36I restriction site.

To generate the ΔMetARIF virus (Fig. 8A, Supplementary Fig. S2A), we constructed a recombinant donor plasmid, pcDNA-ΔMetARIF, using the KOD -Plus- Mutagenesis Kit (TOYOBO, Japan). The pcDNA-arif1 plasmid[23] containing *arif-1* and approximately 3 kbp of flanking sequences of the BmNPV T3 genome was used as the template. Inverse PCR was performed with the primers "arif-1_ΔMet_inv_F" and "arif-1_ΔMet_inv_R" to introduce a deletion of the amino acid region +1 to +220 of ARIF-1. This deletion results in the loss of the first methionine of ARIF-1 (the next methionine is located at amino acid residue +238). The inverse PCR product was self-ligated, amplified in *E. coli* competent cells, sequenced, and used as pcDNA-ΔMetARIF. Subsequently, pcDNA-ΔMetARIF was cotransfected into BmN cells with the Bsu36I-digested genomic DNA of ARIFD. The FuGENE HD transfection reagent (Promega) was used for transfection. ΔMetARIF was isolated by identifying white plaques that did not express β-galactosidase using a plaque assay with agarose overlays containing 5-bromo-4-chloro-3-indolyl-β-D-galactoside[87].

To generate ARIF-GFP and its variants (Fig. 8A, Supplementary Fig. S2A), we introduced a BamHI restriction site just upstream of the *arif-1* stop codon in pcDNA-arif1 using the KOD -Plus- Mutagenesis Kit with the following primers: "arif-1_BamHI_C-inv_F" and "arif-1_C-inv_R." The resultant plasmid (pcDNA-ARIF-BamHI) was then digested with BamHI and ligated with a BamHI-digested PCR fragment of *egfp*. The *egfp* fragment was amplified from the hspGFP genome using the primers "EGFP_BamHI_F" and "EGFP-stop_BamHI_R." This ligation step produced the pcDNA-ARIF-GFP plasmid. Additionally, we performed partial deletions of ARIF-1 using the KOD -Plus- Mutagenesis Kit and the primers listed in Supplementary Table S1. These deletions led to the generation of the following plasmids: pcDNA-ARIFΔEXL-GFP (deletion of the extracellular loop corresponding to the amino acid region +130 to +194), pcDNA-ARIFΔC1-GFP (deletion of the C-terminal amino acid region +346 to +440), pcDNA-ARIFΔC2-GFP (deletion of the C-terminal amino acid region +226 to +440), and pcDNA-ARIFΔC3-GFP (deletion of the C-terminal amino acid region +226 to +345). These plasmids were cotransfected with the Bsu36I-digested genomic DNA of ARIFD into BmN cells to obtain recombinant viruses, followed by the previously described isolation method.

The deletion or recombination of each gene in these viruses was validated by PCR and DNA sequencing. All the primers used in these experiments are listed in Supplementary Table S1.

### Scanning electron microscopy

Virus-infected or mock-infected larvae were excised in PBS to obtain tracheal tissues with attached hemocytes. Two tissues from different individuals in each experimental group were obtained. The tissues were fixed in 2% PFA and 2.5% glutaraldehyde in phosphate buffer (100 mM, pH 7.2) at 4 °C for >O/N until use. The tissues were then washed twice with DW, dehydrated in a graded ethanol series (70%, 90%, 95%, and 100%), and transferred to t-butyl alcohol for two changes. After 5 min freezing in a refrigerator, the samples were dried by the critical point drying device JCPD-5 (JEOL Ltd., Japan), followed by sputter-coating with platinum palladium (E-1030, Hitachi Ltd., Japan). Finally, the samples were examined using an SEM S-4800 (Hitachi Ltd., Japan) at an accelerating voltage of 5.0 kV.

### Quantification of virus production

At 96 hpi, hemolymph was collected from virus-infected fifth-instar larvae, and viral OBs released in the hemolymph were counted using a hemocytometer, as previously reported[88].

### Statistics and reproducibility

The statistical analyses were conducted using Prism 8 software and R (ver. 4.4.0)[85] with the effsize package (ver. 0.8.1) and the dunn.test package (ver. 1.3.6). Mann–Whitney U test was used to compare two groups of data presumed not to follow a normal distribution. Kruskal–Wallis rank sum test was used to compare three or more groups of data presumed not to follow a normal distribution, followed by Dunn's multiple comparisons test with Bonferroni's correction. For data with normal distribution, unpaired *t*-test (two-tailed) (two groups) and one-way ANOVA (three or more groups) were used. One-way ANOVA was followed by Tukey's multiple comparisons test. Chi-square test was used to analyze count data. Log-rank (Mantel-Cox) test with Bonferroni's correction was used to analyze survival curves. The effect sizes (Cohen's *d* for unpaired *t*-test and *r* for other tests) were calculated using R. Detailed information regarding the sample sizes, the number of replicates, and the specific statistical methods and parameters employed is provided in the figure legends, the "Methods" section, or Supplementary Data.

### Reporting summary

Further information on research design is available in the Nature Portfolio Reporting Summary linked to this article.

### Data availability

Source data for manuscript graphs and images are available in Supplementary Data (excel file).

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

## Acknowledgements

We thank M. Fujimoto, H. Takanashi, S. Arimura, and N. Tsutsumi (Univ. Tokyo) for confocal microscope observation including technical lecture and suggestion, K. Tomita (Technology Advancement Center, Graduate School of Agricultural and Life Sciences, Univ. Tokyo) for scanning electron microscopy observation including technical lecture and suggestion, L. E. Volkman (Univ. California, Berkeley) for providing anti-VP39 antibody, R. Sato (Tokyo Univ. Agri. Technol.) for providing anti-Hemocytin antiserum, Division of Biological Science and Technology, Graduate School of Natural Science and Technology, Kanazawa Univ. for the use of a confocal microscope, T. Kiya (Kanazawa Univ.) for providing some reagents for immunostaining and confocal microscope observation, H. Endo (RIKEN) for helpful advice about methods for suppressing the immune response, H. Hikida (Kyoto Univ.) for rearing insects, and M. Kawamoto (Univ. Tokyo) for clerical assistance. R.K. is a recipient of a fellowship from the Japan Society for the Promotion of Science (DC1 and PD). This work was supported by Japan Society for the Promotion of Science (JSPS) Grants-in-Aid for Scientific Reseaerch (B) (25292196, 16H05051, 19H02966) to S.K., Grant-in-Aid for Transformative Research Areas (A) "Co-evolutionary emergence of extended phenotypes: Elucidation of the molecular mechanisms of extended phenotypes" (24H02290) to S.K., Grant-in-Aid for Research Activity Start-up (15H06155) to R.K., Grants-in-Aid for JSPS Fellows (12J06034, 18J00134) to R.K., Grants-in-Aid for Early-Career Scientists (21K14860, 24K08930) to R.K.

## Author contributions

Conceptualization: R.K. Methodology: R.K. Investigation: R.K. Visualization: R.K. Funding acquisition: R.K., S.K. Project administration: R.K., S.K. Supervision: S.K. Writing—original draft: R.K. Writing—review & editing: S.K.

## Competing interests

The authors declare no competing interests.
