## [Transparent Peer Review file · Communications Biology]

Baculoviruses remodel the cytoskeleton of insect hemocytes to breach the host basal lamina

Corresponding Author: Dr Ryuhei Kokusho

Version 0:

Reviewer comments:

Reviewer #1

(Remarks to the Author)

Review of: Baculovirus remodels the cytoskeleton of silkworm hemocytes for breaching the basal laminal barrier

Overview:

The authors present interesting findings about a baculovirus protein (ARIF-1), expressed in infected hemocytes, in promoting transit through basal laminae during BmNPV secondary infection. Many prior studies have focussed on discovering how viruses breach the midgut barrier, going from the midgut lumen through gut epithelia and the basal lamina, into the insect circulatory system (hemocoel). However, this study looks at how viruses in the hemocoel bypass basal lamina to enable them to infect secondary tissues of the host caterpillar. The baculovirus protein of interest to this study (and the described phenomena) is ARIF-1 which has already been documented as important for breaching the midgut by a similar molecular means – rearrangement of cellular actin and formation of filopodia (invadosomes) that invade tissues by directing enzymatic dissolution of the basal lamina. The authors describe their findings within the context of prior studies showing baculoviruses escape the gut by infecting cells of airways (tracheae) that then act as conduits for systemic spread.

General comments:

The use of the hspGFP reporter was useful to track infection after hemocoelic injection. The lack of ARIF-1 expression clearly had effects on how quick it infected trachea – wt virus seemed about 20h ahead of ARIFD (Fig1), and this is corroborated by much lower levels of fat body infection foci for the ARIFD mutant (Fig2). They then assess infected hemocytes' ARIF-1-dependent localization on internal tissues (Fig3) and cellular sub-structures (filopodia/invadosomes) using an ARIF-GFP fusion reporter (Fig4). Some of the microscopy images are less than convincing (see specific comments) while other images are very representative of their conclusions. Overall an interesting study that I have not seen considered before (navigating through BL to infect secondary tissues) but that could be improved by some additional and/or better immunostaining or other data (which are suggested in the specific comments). I think the nature of the study is of high interest to baculovirologists and perhaps arbovirologists interested in virus dissemination in animal tissues.

Specific comments:

- 1-It would be more demonstrative in Figure 2 if the authors were point out both the large and small infection foci. It is clear they are all small for ARIFD, but it is hard to discern the large and small foci they describe for wt virus infected fat bodies.
- 2-The microscopy in Figure 3 is not very good. I realize these are immunostained tissues that have a lot of autofluorescence, but its very difficult to fully discern the cells/tissues and immunostain signals – the outlines are helpful but I do not think the images are good enough to make the claim that infected hemocytes attach to tracheoblasts.
- 3-The y-axis scale on the Figure 3E chart should be adjusted to show the reality of how different the hemocyte counts were; the statistics should verify differences in counts between the wt and ARIFD viruses, but the way the graph is made it looks a bit too different than 2^6 vs 5^6 .
- 4-The microscopy in Figure 4 is far superior to that in Figure3, indicating that GFP-labelled hemocytes can be better imaged. The overlay of actin and GFP-ARIF signals is quite clear from separate images but less so in the merged image. If they authors wish to claim ARIF/actin colocalization they should provide better merge images and/or mathematical colocalization analysis (Pearsons?).
- 5-the images (Fig 4E) of the truncated ARIF-GFP in hemocytes suffers from the same poor microscopy as noted for Fig3. Is this because the GFP-ARIF proteins are not well expressed (ie folding problems)? If the truncated fusion proteins do not express/fold well, then they will not traffic as well to the cell membrane and thus reduce function...but not per se because the domain is needed but simply because of mis-folding. This matter of protein stability could be addressed by immunoblotting to discern relative levels of total protein between wt GFP-ARIF and its truncations.
- 6-It is nice to see the difference in hemolymph OB counts at 96 h for wt and the ARIF-mutant viruses, but in light of the

“delayed” infection described by the authors (in a prior study) by ARIFD BmNPV it would be even better to compare OB levels in terminal larval carcasses.

7-The images and tracing of signal overlap (or lack of) between ARIF and COLIV, filopodia and COLIV, CAPSIDs inside filopodia and between GP64 and ARIF is impressive, but I wonder how many times they have observed these overlap? I know its difficult to combine data of replicated experiments, but if the authors indicated how many times they attempted and found similar results from this expt it would strengthen their case.

8-The anti-VP39-immunostained capsids in nuclei devoid of an overwhelming soluble VP39 signal is an interesting observation (that capsids were delivered by the attached hemocyte), but the image is not so clear and besides is it steadfastly held that baculovirus capsids enter the nucleus rather than uncoating at nuclear pores? The authors should adjust their conclusions on this data with what is accepted in the literature for alphabaculoviruses (or at least BmNPV).

Reviewer #2

(Remarks to the Author)

In this manuscript, Kokusho et al. attempt to elucidate the role of the baculovirus Arif-1 in establishing systemic infection. They revealed that Arif-1 induces the remodeling of F-actin structures in hemocytes, forming filopodia-like protrusions and invadosome-like structures that penetrate the basal lamina barrier, facilitating the budded virions spread to the trachea or other tissues. This manuscript introduces a previously unreported mechanism for cell-to-cell spread in baculovirus via filopodia and invadosomes.

Major comments:

1. Many viruses can be observed to mediate cell-to-cell spread through filopodia in cultured cells in vitro. It has been found that the baculovirus Arif-1 can form invadosome structures in cultured insect cells, but can it also form filopodia structures? If the authors could demonstrate the formation of filopodia structures and the transmission of the virus through filopodia structures in vitro, it would lend strong support to their in vivo findings.
2. The results in Fig. 5 seem to be contradictory. The results in Figs. 5A-C suggest that the invadosome structures penetrate the BL barrier, while the filopodia do not insert into the BL. However, in Fig. 5E, the authors present results showing the colocalization of the viral capsid protein VP39 with the filopodia structures. Therefore, it is difficult to draw the conclusion that the virus employs filopodia or invadosome structures for cell-to-cell spread based on these results. More evidence is still required.
3. BmNPV infection in BmN cells leads to the vanishing of microvilli, cilia, and lamellipodia on the cellular surface, which are replaced by a substantial presence of vesicular structures. The evidence provided in this manuscript indicates that virus-infected hemocytes form filopodia structures. This observation differs from prior findings in BmN cells. The authors should, therefore, employ SEM to further elucidate the impact of viral infection on the morphology of the hemocyte surface to further substantiate their findings.
4. At 24 h p.i., the infection foci in fat body infected by the Arif-1 mutant virus exhibit reduced dimensions compared to those infected with the wild-type virus (Fig. 2A). Considering that at 24 hours the virus has just completed one replication cycle and begun to infect surrounding cells, it is difficult to attribute the difference in foci size to the transmission function of Arif-1. This appears to be a consequence of differential infectious virulence between these two viruses. Additionally, given that Arif-1 is not a viral structural protein, its role in modulating infectious virulence upon initial infection remains unclear. The authors are encouraged to extend their observations to the later stages of infection to more accurately assess the influence of the Arif-1 mutation on viral propagation and foci expanding.

Minor comments:

1. Due to the intrinsic autofluorescence of silkworm tissues, reliance on fluorescence imaging alone may be inadequate. It is suggested that the authors add quantitative data to demonstrate the differences between wild-type and mutant viruses' infection more effectively. Such as genome copies or GFP expression levels.
2. Why is the host hsp70 promoter used to drive the expression of the reporter gene? Have the authors confirmed whether mutation in Arif-1 affect the expression of the hsp70 promoter?
3. The authors should provide replication and proliferation data for several Arif-1 mutant viruses to exclude the possibility that Arif-1 promotes systemic infection by participating in viral replication and proliferation.
4. What does Fig. 5D aim to illustrate?

Reviewer #3

(Remarks to the Author)

Baculoviruses are important insect pathogens and have also been broadly used as expression vectors. Studies dating to the 1980s have examined the mechanisms underlying how baculoviruses systemically infect insects with evidence implicating both the host tracheal system and hemocytes in disseminating the budded form of baculoviruses to different tissues. This study advances results reported by these authors in 2015 (JGV) on an alphabaculovirus gene named actin rearrangement-inducing factor 1 (arif-1). These earlier studies indicated loss of ARIF-1 in BmNPV slowed systemic infection of *Bombyx mori* larvae. This study used a GFP-expressing recombinant BmNPV together with or without a functional arif-1 gene to further advance understanding of ARIF-1 functions. The authors report that BmNPV infects *B. mori* immune cells (hemocytes), which causes ARIF-1 induced alterations to the cytoskeleton that promote the formation of filopodia and an invadosome-like

structure that promotes the entry of budded virus through the basal lamina and into tracheal cells. The authors further present microscopy-based evidence indicating budded then moves from trachea to other tissues like the fat body.

My overall assessment is that the study presents new, information on how alphabaculoviruses systemically infect hosts that will be of interest to both virologists and immunologists that study arthropods. While prior studies emphasize entry of BVs from the gut to tracheoblasts, this study shows that BVs can also be transferred from hemocytes to tracheal epithelial cells. Together with another recently published study (Lauko et al. 2021 MBC), this paper will also likely be of interest to the broader cell biology community as it shows that baculovirus-induced invadosomes show features that are similar to the invadosomes some vertebrate pathogens produce. The paper is overall well organized, clearly written, and the results overall support main conclusions. I have only a small number of suggestions for revision.

1. A general but important omission from the paper in my opinion is the growing literature indicating a substantial number of hemocytes in some insects are intrinsically sessile. Some sessile hemocytes proliferate and thus contribute to the population of cells in circulation and also can dynamically change the proportion of cells in circulation. Most of this literature has accrued from studies of *Drosophila* and mosquitoes with relatively little literature in Lepidoptera (like *B. mori*). Finding these studies is also easily done by just key wording 'sessile hemocytes AND insect' in a general literature search. Despite the bias of the current literature toward Diptera, there are strong reasons to suspect that sessile hemocytes are a common phenomenon in insects generally including moth larvae. This is important to this study for obvious reasons as it would suggest the possibility baculovirus infection alters how many or few sessile hemocytes are present as opposed to fundamentally changing hemocytes from being exclusively in circulation. I thus would strongly suggest the authors bring this possibility into the discussion section of the manuscript and at least cite a couple of papers indicating the types of hemocytes that have been identified to bind to the surface of different tissues in the absence of virus infection and possible implications for the results reported in this study.

2. Along these lines, the authors mention the possible roles of fgf in hemocyte migration and how this might fit with BmNPV encoding an fgf gene that could be relevant to interactions with trachea. That's fine but given the broader literature, I would be suspicious *B. mori* hemocytes preferentially bind to only trachea when very careful studies in other insects present evidence sessile, granulocyte-like cells bind to a number of other tissues that are also enveloped by the basal lamina. The authors make no mention of whether they restricted their microscopy studies to only looking at trachea or alternatively potentially observed hemocytes (whether infected or not) on the surface of other tissues. I would suggest at that the authors at least generally address this point as either a key future need or that the possibly did see hemocytes on other tissues but didn't pursue this in the current study.

I fully recognize the authors' interest in the tracheal system is driven primarily by the baculovirus literature and prior studies that implicate it in promoting systemic infection. On the other hand, given more recent evidence describing sessile hemocytes on the surface of several tissues in insects, I cannot intuitively see why ARIF-1's role in promoting the formation of filopodia that penetrate the basal lamina should necessarily be restricted to the tracheal system given most tissues in the hemocoel are also covered by the basal lamina. In turn, this could suggest infected hemocytes could use invadosomes to penetrate the basal lamina to infect other cell types besides tracheal epithelia. If the authors have results indicating a real bias that infected granulocytes preferentially associate themselves with trachea, they should report it. If they don't, they should again at acknowledge this is an unknown that needs to be examined in the future.

3. Fig. 3. I would think an important graph to show in addition to the number of pH3+ hemocytes (Fig. 3D) and total number of hemocytes in circulation (Fig. 3E) would be the proportion of these cells that were BmNPV infected as evidenced by detection of GFP. Moreover, are these hemocytes predominantly granulocytes, which is the only hemocyte type the authors mention in the study as being infected, or are similar proportions of other hemocyte types in circulation (plasmatocytes, spherule cells, oenocytoids) also BmNPV infected (which I would suspect). If the latter, a possible future direction would also be if ARIF-1 only induces the formation of invadosomes in infected granulocytes or cytoskeletal rearrangements result in other hemocyte types also producing these structures (at minimum plasmatocytes which are also a strongly adhesive in *B. mori* and other Lepidoptera).

4. I fully understand why the authors conducted their experiments by injecting budded virus into the hemocoel as oral infection would create confounding conditions that could make interpretation of the data presented. Still, injecting virus directly into the hemocoel is also 'non-natural'. Thus, it could be valuable/important for the authors to show or at least mention from other studies that per os infected *B. mori* also results in early infection of both hemocytes and tracheoblasts and that hemocytes projecting invadosomes into tracheal epithelial cells occurs. This could easily be looked at if the authors haven't already done so, and I think should be shown to make clear the results reported occur after normal infection.

Minor points

1. While the authors do a good job of reporting N values in the graphs presented, I didn't get a clear sense of the number of larvae or samples the authors examined to generate the microscopy data, and in essence how common or rare the granulocytes in association with trachea were as shown in Fig. 3-5. Some mention of sample sizes in the Methods or elsewhere for this aspect of the study would be helpful.

2. As noted above, the paper is overall well written but I did see a few sentences/phrasing that was a bit awkward. On the first page of the Introduction, for example, the authors use the term 'open blood-vascular system' (line 37) but this isn't wording that insect immunologists normally use. Instead, correct phrasing would simply be 'an open circulatory system'. Line 38, should be changed to 'with a basal lamina' from 'with the basal lamina'. Line 46, insect-related should be revised to

insect- associated. Select other pages have similar minor grammatical/phrasing errors. Potentially just rereading carefully would be enough to catch these and correct them.

Version 1:

Reviewer comments:

Reviewer #1

(Remarks to the Author)

Review comments: (related to my specific concerns of the initial review)

1-it is much easier to discern the large vs small foci, and the quantification (Fig2E) really supports the claim.

2-the hemocyte staining is much better and is now quite convincing

3-the adjusted graph scaling looks good. However in Fig 8 it is still a bit deceiving showing mutant viral OB levels near zero, but WT OBs at 10^9 . Since authors do indeed state in the results that it's a 30-fold reduction for the mutants, it makes it clear the difference but also highlights the issue I have with the zero straight to 10^9 . It's a minor thing and I don't think it needs any adjustment (I have the same comment for the graph in fig 3F).

4-the microscopy images are well improved, and the additional immunoblot analyses are a nice addition...but they do show there is much less of the dC1 and dC2 proteins (and a lot more of the dC3 protein), perhaps indicating maturation problems, but I do agree there is enough detectable in the immunostained cells.

5-The terminal carcass OB levels, in each gram weight of tissue, is much more convincing data.

An outstanding issue to resolve:

One issue that needs to be confirmed/addressed is that the authors state (in fig 2E, lines 146 and 651) that the size of the "large" infection foci is 1000 mm². When I did a quick online conversion of area units, it equated 1000 mm² to about 1.5 inches², and so I think this sizing may be at least an order of magnitude off (more like 100 mm²?).

Overall, I think this is a very interesting and convincing set of experiments and that the authors' statements reflect the data well.

Reviewer #2

(Remarks to the Author)

The author has thoroughly addressed the issues I raised in current revised manuscript. In particular, the scanning electron microscopy results and the findings regarding filopodia-like protrusions and vp39 localization have significantly improved the completeness and credibility of this work. This study will greatly advance our understanding of the mechanisms underlying the rapid spread of baculoviruses within the host. I have no additional comments.

Reviewer #3

(Remarks to the Author)

I was Reviewer #3 of the first submission. I read the revised manuscript and thought the authors did an excellent job addressing my prior concerns. It also appeared they did a very good job of carefully and thoughtfully addressing the concerns of the other two reviewers.

In my previous review, I also pointed out a handful of grammatical/syntax errors. The authors corrected these and asked that I specify any other suggested corrections I might see. I have done this below. I did not previously mention this, but I also suggest the authors be a little more precise about arif-1 only being found in certain baculoviruses (not all), which is another feature that is likely important for function.

Abstract and Introduction--host's body fluid: host hemocoel (or host's circulatory system)

Introduction

Baculovirus is: Should be Baculoviruses are

Next, baculoviruses need to cross the midgut BL to establish infection in other tissues. Correct to 'Next, most baculoviruses cross the midgut BL...'.
'.

I suggest this because some baculoviruses only infect the midgut of hosts and do not establish systemic infections. The authors should also state either in the Introduction, Discussion or both that arif-1 isn't a baculovirus core gene. I did not take the time to look carefully myself, but George Rohrmann's 2019 review reports that arif-1 is present in most/all alphabaculoviruses (Group I and II) but is apparently absent in betabaculoviruses (GVs) (which also systemically infect hosts), or gamma- and deltabaculoviruses (which primarily or exclusively infect only the midgut). Thus, my overall suggestion is for the authors not use the term 'baculovirus' when referring to arif-1 and more precisely refer to its presence in alphabaculoviruses (=nucleopolyhedroviruses (NPVs) which infect Lepidoptera.

Alternatively, baculoviruses utilize the host trachea as an escape route from the midgut (63 Supplementary Fig. S1A, [iii-b])

(14). Revise slightly by adding the word 'most' before baculoviruses' (for the same reason as noted above).

In this study, we discovered that the baculovirus membrane protein ARIF-1 (actin rearrangement-inducing factor 1) modifies the actin cytoskeleton of host hemocytes and enables them to breach the BL barrier, establishing a highly efficient systemic infection.

Modify for precision to: In this study, we discovered that the membrane protein ARIF-1 (actin rearrangement-inducing factor 1) of *Bombyx mori* nucleopolyhedrovirus modifies the actin cytoskeleton of host hemocytes and enables them to breach the BL barrier, establishing a highly efficient systemic infection.

Results

First subheading. Baculovirus can directly establish infection from the hemolymph at the central tracheal region despite the BL barrier'

Would suggest the authors revise this subheading to 'Bombyx mori nucleopolyhedrovirus can directly establish infection from the hemolymph at the central tracheal region despite the BL barrier' (again, because not all baculoviruses systemically infect hosts).

Line 104. Change viral arif-1 gene to BmNPV arif-1 gene.

Line 159. Baculovirus-infected hemocytes attach to the invasive infection foci in the trachea

Revise to: BmNPV- infected hemocytes attach to the invasive infection foci in the trachea

Line 181. disappeared from the hemolymph by BmNPV infection. Revise to 'disappeared from the hemolymph after BmNPV infection'.

Line 321-325. In summary, our results suggest that baculoviruses remodel host hemocytes to form ARIF-1-derived structures for attaching to the tissue surface and then breach the BL barrier to deliver progeny virions directly into the tissues. This ARIF-1-mediated hemocyte remodeling greatly contributes to the quick spread of virus infection throughout the host body.

Revise to: In summary, our results suggest that BmNPV remodels host hemocytes to form ARIF-1-derived structures for attaching to the tissue surface and then breach the BL barrier to deliver progeny virions directly into the tissues. This ARIF-1-mediated hemocyte remodeling greatly contributes to the quick spread of virus infection throughout the hos.

Discussion

ARIF-1 has been reported as a baculoviral inducer of actin rearrangement in lepidopteran cultured cells (17).

Revise to: ARIF-1 has been reported as an inducer of actin rearrangement in lepidopteran cultured cells after BmNPV infection (17).

Line 506. Since baculoviruses highly express arif-1 during the initial stage of midgut infection (71), ARIF-1-mediated cytoskeletal modification may also occur in the midgut.

Modify to 'Since baculoviruses like BmNPV highly express arif-1 during the initial stage of midgut infection (71), ARIF-1-mediated cytoskeletal modification may also occur in the midgut.

Currently, one plausible theory of the midgut escape is that baculoviruses overcome the midgut BL barrier by escaping to the hemocoel through tracheoblast infection (14), as described in the Introduction section and Supplementary Fig. S1.

Again, suggest revision to: 'Currently, one plausible theory of the midgut escape is that BmNPV and other alphabaculoviruses like AcMNPV overcome the midgut BL barrier by escaping to the hemocoel through tracheoblast infection (14), as described in the Introduction section and Supplementary Fig. S1.

Following up on this, the authors maybe should also add a followup sentence indicating the above 'plausible' theory potentially applies to most or all alphabaculoviruses (NPVs) but not other baculoviruses given the apparent absence of arif-1 genes in betabaculoviruses which also escape the midgut. On the other hand the absence of arif-1 in delta and gammabaculoviruses potentially contributes to viruses in these genera infecting primarily or exclusively the midgut of hosts.

Reviewer #1

We thank reviewer #1 for the valuable and helpful comments.

All changes are indicated by using the Track Changes function of Microsoft Word. Please see the Word file of our article to see the changes indicated below (the line numbers of the article pdf file available in the editorial system may differ from those of the original Word file).

Overview:

The authors present interesting findings about a baculovirus protein (ARIF-1), expressed in infected hemocytes, in promoting transit through basal laminae during BmNPV secondary infection. Many prior studies have focussed on discovering how viruses breach the midgut barrier, going from the midgut lumen through gut epithelia and the basal lamina, into the insect circulatory system (hemocoel). However, this study looks at how viruses in the hemocoel bypass basal lamina to enable them to infect secondary tissues of the host caterpillar. The baculovirus protein of interest to this study (and the described phenomena) is ARIF-1 which has already been documented as important for breaching the midgut by a similar molecular means – rearrangement of cellular actin and formation of filopodia (invadosomes) that invade tissues by directing enzymatic dissolution of the basal lamina. The authors describe their findings within the context of prior studies showing baculoviruses escape the gut by infecting cells of airways (tracheae) that then act as conduits for systemic spread.

General comments:

The use of the hspGFP reporter was useful to track infection after hemocoelic injection. The lack of ARIF-1 expression clearly had effects on how quick it infected trachea – wt virus seemed about 20h ahead of ARIFD (Fig1), and this is corroborated by much lower levels of fat body infection foci for the ARIFD mutant (Fig2). They then assess infected hemocytes' ARIF-1-dependent localization on internal tissues (Fig3) and cellular sub-structures (filopodia/invadosomes) using an ARIF-GFP fusion reporter (Fig4). Some of the microscopy images are less than convincing (see specific comments) while other images are very representative of their conclusions. Overall an interesting study that I have not seen considered before (navigating through BL to infect secondary tissues) but that could be improved by some additional and/or better immunostaining or other data

(which are suggested in the specific comments). I think the nature of the study is of high interest to baculovirologists and perhaps arbovirologists interested in virus dissemination in animal tissues.

(Authors' reply)

We thank this reviewer for their encouraging comments.

Specific comments:

1-It would be more demonstrative in Figure 2 if the authors were point out both the large and small infection foci. It is clear they are all small for ARIFD, but it is hard to discern the large and small foci they describe for wt virus infected fat bodies.

(Authors' reply)

Thank you for your constructive comment. We replaced the images of hspGFP-infected fat bodies (Fig. 2A, upper panels) with more demonstrative ones to distinguish large and small infection foci more easily. Also, we calculated the percentage of infection foci over and under 1000 μm^2 and found a significant difference between the viruses by Chi-square test (Fig. 2E of the revised manuscript). We revised the Results and Discussion sections according to these results (lines 139–177 and 824–834 of the revised manuscript).

2-The microscopy in Figure 3 is not very good. I realize these are immunostained tissues that have a lot of autofluorescence, but its very difficult to fully discern the cells/tissues and immunostain signals – the outlines are helpful but I do not think the images are good enough to make the claim that infected hemocytes attach to tracheoblasts.

(Authors' reply)

Thank you for your constructive comment. The images in Fig. 3 are photos with a maximum magnification of our zoom microscopy (Axio Zoom.V16 with 2.3x objective lens (ZEISS)); therefore, it is difficult to increase the resolution using this zoom microscopy. Instead, we added new confocal microscopic images clearly showing the hemocyte attachment on the tracheal surface (Figs. 3A–B of the revised manuscript).

3-The y-axis scale on the Figure 3E chart should be adjusted to show the reality of how different the hemocyte counts were; the statistics should verify differences in counts between the wt and ARIFD viruses, but the way the graph is made it looks a bit too different than 2^6 vs 5^6 .

(Authors' reply)

Thank you for your comment. The y-axis in Fig. 3E (Fig. 3F of the revised manuscript) is linear, and the y-axis scales are 0, 2×10^6 , 4×10^6 , 6×10^6 , and 8×10^6 from the bottom. The mean values of hspGFP and hspGFP-ARIFD are about 2×10^6 and 5×10^6 , respectively. Thus, this 2.5-fold change is correctly reflected in the y-axis of Fig. 3E. Please let me know if the y-axis scales are not displaying correctly in your file or if I misunderstood what you meant by your comment.

4-The microscopy in Figure 4 is far superior to that in Figure3, indicating that GFP-labelled hemocytes can be better imaged. The overlay of actin and GFP-ARIF signals is quite clear from separate images but less so in the merged image. If they authors wish to claim ARIF/actin colocalization they should provide better merge images and/or mathematical colocalization analysis (Pearsons?).

(Authors' reply)

Thank you for your constructive suggestions. We obtained additional confocal images of ARIF-1 protrusions and analyzed colocalization using the Coloc 2 plugin in Fiji. The mean Pearson's colocalization coefficients of ARIF-1 and F-Actin in filopodia-like protrusions and invadosome-like structures are $r = 0.516$ and 0.598 , respectively, indicating that ARIF-1 and F-Actin are moderately colocalized (Fig. 4F of the revised manuscript). We added a description of this point in the Results section (lines 236–238 and 263–265 of the revised manuscript).

5-the images (Fig 4E) of the truncated ARIF-GFP in hemocytes suffers from the same poor microscopy as noted for Fig3. Is this because the GFP-ARIF proteins are not well expressed (ie folding problems)? If the truncated fusion proteins do not express/fold well, then they will not traffic as well to the cell membrane and thus reduce function...but not per se because the domain is needed but simply because of mis-folding. This matter of protein stability could be addressed by immunoblotting to discern relative levels of total protein between wt GFP-ARIF and its truncations.

(Authors' reply)

Thank you for your constructive comment. As mentioned above (in reply to your comment #2), the poor quality of the microscopic images is due to the limitation of our zoom microscopy. We replaced the images in Fig. 4E with high-resolution confocal

microscopic images (Fig. 8B in the revised manuscript), which clearly show the cell morphology of hemocytes infected with ARIF-GFP variants. Also, we performed immunoblot analysis of cultured cells (BmN-4 cells) infected with ARIF-GFP mutants. The expression of some of the truncated ARIF-GFP proteins was slightly lower than that of full-length ARIF-GFP (Supplementary Fig. S8), whereas we were able to observe proper membrane localization of truncated ARIF-GFP proteins in cultured cells (Supplementary Fig. S9). These results suggest that the expression of partially deleted ARIF-GFP variants is high enough to observe cell morphology. We revised the Results section according to these results (lines 425–426 of the revised manuscript).

6-It is nice to see the difference in hemolymph OB counts at 96 h for wt and the ARIF-mutant viruses, but in light of the “delayed” infection described by the authors (in a prior study) by ARIFD BmNPV it would be even better to compare OB levels in terminal larval carcasses.

(Authors’ reply)

Thank you for your suggestion. In this revision, we found that the deletion of ARIF-1 caused a 25% increase in OB levels in terminal larval carcasses (Supplementary Fig. S11A), suggesting that the delay of infection in hspGFP-ARIFD ultimately caught up with and surpassed that of hspGFP. This 25% increase might be partially due to the increase in body weight, although statistically not significant (Supplementary Fig. S11B)—hspGFP-ARIFD-infected larvae continued feeding until the later stage of infection because of the delay of systemic infection. In the case of partial deletion of ARIF-GFP, some mutant viruses showed reduced or increased BV production examined at 24 hpi (Supplementary Fig. S10B), presumably affecting the OB levels in the larval carcasses. Therefore, we focus on the morphology of virus-infected hemocytes. We described this point in the Discussion section (lines 777–779 of the revised manuscript).

7-The images and tracing of signal overlap (or lack of) between ARIF and COLIV, filopodia and COLIV, CAPSIDs inside filopodia and between GP64 and ARIF is impressive, but I wonder how many times they have observed these overlap? I know its difficult to combine data of replicated experiments, but if the authors indicated how many times they attempted and found similar results from this expt it would strengthen their case.

(Authors' reply)

Thank you for your constructive suggestion. In this revision, we calculated the ratio of the formation of ARIF-1-derived structures (Fig. 4G), their BL invasion (Fig. 6F), nucleocapsid overlap (Fig. 7D), and nucleocapsid invasion into tissues (Fig. 7E), based on the confocal observation of over 20 hemocytes. Also, we described the number of tracheal samples observed in the Methods section (lines 1124–1130 of the revised manuscript).

8-The anti-VP39-immunostained capsids in nuclei devoid of an overwhelming soluble VP39 signal is an interesting observation (that capsids were delivered by the attached hemocyte), but the image is not so clear and besides is it steadfastly held that baculovirus capsids enter the nucleus rather than uncoating at nuclear pores? The authors should adjust their conclusions on this data with what is accepted in the literature for alphabaculoviruses (or at least BmNPV).

(Authors' reply)

Thank you for your comment. We added the higher-resolution confocal images and their 3D reconstruction data in Fig. 7C and Supplementary Fig. S9 of the revised manuscript; these data clearly show dot-like VP39 signals (presumably nucleocapsids) in the nucleus of tracheal epithelial cells. As reviewed in Blissard & Theilmann (*Annu. Rev. Virol.*, 2018, doi:10.1146/annurev-virology-092917-043356), nucleocapsids of AcMNPV (a type species of *Alphabaculovirus* that is closely related to BmNPV) can enter the nucleus without capsid disassembly (e.g., Ohkawa et al., *J. Cell Biol.*, 2010, doi:10.1083/jcb.201001162). Our observation suggests that BmNPV employs the same nuclear entry mechanism as AcMNPV. We have added the description in the Results section (lines 402–405 of the revised manuscript).

Thank you again, reviewer #1, for your valuable and constructive comments.

Reviewer #2

We thank reviewer #2 for the valuable and helpful comments.

All changes are indicated by using the Track Changes function of Microsoft Word. Please see the Word file of our article to see the changes indicated below (the line numbers of the article pdf file available in the editorial system may differ from those of the original Word file).

In this manuscript, Kokusho et al. attempt to elucidate the role of the baculovirus Arif-1 in establishing systemic infection. They revealed that Arif-1 induces the remodeling of F-actin structures in hemocytes, forming filopodia-like protrusions and invadosome-like structures that penetrate the basal lamina barrier, facilitating the budded virions spread to the trachea or other tissues. This manuscript introduces a previously unreported mechanism for cell-to-cell spread in baculovirus via filopodia and invadosomes.

(Authors' reply)

We thank this reviewer for their encouraging comments.

Major comments:

1. Many viruses can be observed to mediate cell-to-cell spread through filopodia in cultured cells *in vitro*. It has been found that the baculovirus Arif-1 can form invadosome structures in cultured insect cells, but can it also form filopodia structures? If the authors could demonstrate the formation of filopodia structures and the transmission of the virus through filopodia structures *in vitro*, it would lend strong support to their *in vivo* findings.

(Authors' reply)

Thank you for your constructive suggestion. We agree with your comment on the importance of *in vitro* assessment of filopodia formation and filopodia-mediated viral transmission. However, it is technically challenging to evaluate the effect of ARIF-1 on filopodia formation because our cultured cells (BmN-4 cells) are adherent and form a large number of filopodia even in uninfected conditions. Also, BmNPV-infected cultured cells become round shape before 24 hpi (the onset of BV egress), which makes the filopodia quantification more challenging. In addition, in contrast to the prior study of invadosome formation by AcMNPV and BmNPV ARIF-1 proteins in cultured cells (Lauko et al., *Mol. Biol. Cell*, 2021, doi:10.1091/mbc.E20-11-0705), invadosome formation is not observed in BmN-4 cells in our experimental condition. Therefore, we

need to find cell lines or experimental conditions suitable for the *in vitro* functional analysis of BmNPV ARIF-1, which we think is the future work of our research. We have added some description about virion transmission in filopodia in the Discussion section to support our *in vivo* findings (lines 559–561 of the revised manuscript).

2. The results in Fig. 5 seem to be contradictory. The results in Figs. 5A–C suggest that the invadosome structures penetrate the BL barrier, while the filopodia do not insert into the BL. However, in Fig. 5E, the authors present results showing the colocalization of the viral capsid protein VP39 with the filopodia structures. Therefore, it is difficult to draw the conclusion that the virus employs filopodia or invadosome structures for cell-to-cell spread based on these results. More evidence is still required.

(Authors' reply)

Thank you for your constructive comment. By additional confocal microscopic observations, we were able to observe the BL penetration of filopodia-like protrusions (Figs. 6A–B) and nucleocapsid transmission in invadosome-like structures (Figs. 7B, D). We think these additional results are enough to corroborate our working model (Fig. 9 of the revised manuscript).

3. BmNPV infection in BmN cells leads to the vanishing of microvilli, cilia, and lamellipodia on the cellular surface, which are replaced by a substantial presence of vesicular structures. The evidence provided in this manuscript indicates that virus-infected hemocytes form filopodia structures. This observation differs from prior findings in BmN cells. The authors should, therefore, employ SEM to further elucidate the impact of viral infection on the morphology of the hemocyte surface to further substantiate their findings.

(Authors' reply)

Thank you for your constructive suggestion. We think filopodia-like protrusions and invadosome-like structures by ARIF-1 are cell type-specific phenomena because previous *in vitro* studies using different cells reported different types of actin remodeling (Kokusho et al., *J. Gen. Virol.*, 2015, doi:10.1099/vir.0.000130; Lauko et al., *Mol. Biol. Cell*, 2021, doi:10.1091/mbc.E20-11-0705; Roncarati & Knebel-Mörsdorf, *J. Virol.*, 1997, doi:10.1128/JVI.71.10.7933-7941.1997; Steffen et al., *Microbiol. Spectr.*, 2023, doi:10.1128/spectrum.05189-22). In this revision, our SEM analysis found that hspGFP-

infected hemocytes not only formed filopodia-like protrusions but also showed a fine mesh-like structure extending widely over the surface of the trachea (Figs. 5B, B' of the revised manuscript). Such structure was not observed in mock- and hspGFP-ARIFD-infected hemocytes (Fig. 5A, C of the revised manuscript), suggesting that ARIF-1 induces this structure. We added the discussion on this point in the Discussion section (lines 797–803 of the revised manuscript).

4. At 24 h p.i., the infection foci in fat body infected by the Arif-1 mutant virus exhibit reduced dimensions compared to those infected with the wild-type virus (Fig. 2A). Considering that at 24 hours the virus has just completed one replication cycle and begun to infect surrounding cells, it is difficult to attribute the difference in foci size to the transmission function of Arif-1. This appears to be a consequence of differential infectious virulence between these two viruses. Additionally, given that Arif-1 is not a viral structural protein, its role in modulating infectious virulence upon initial infection remains unclear. The authors are encouraged to extend their observations to the later stages of infection to more accurately assess the influence of the Arif-1 mutation on viral propagation and foci expanding.

(Authors' reply)

Thank you for your constructive comment. We extended our observations in the fat body to the later stages of infection. We found that the infection of hspGFP-ARIFD spread along the tracheal epithelia or showed patchy infection in the fat body at 36 hpi (Fig. 2B of the revised manuscript). This spreading pattern was different from large infection foci observed in the hspGFP-infected fat body at 24 hpi, which spread in a circular pattern around the tracheal branch points (Fig. 2A). These results suggest that ARIF-1-mediated large infection foci are due to the different mechanism from the tracheoblast-mediated spread (lines 172–177 of the revised manuscript).

At present, we do not think the difference in foci size is caused by the difference in viral infectivity, although further analysis is needed for verification. This is because ARIF-1 is not considered a viral structural protein (as you pointed out). Also, even if ARIF-1 contributes to BV infectivity, this will probably increase the number of infection foci rather than their size at 24 hpi. We observed the BL penetration and nucleocapsid delivery by both filopodia-like protrusions and invadosome-like structures (Figs. 6, 7 of the revised manuscript). This result suggests direct nucleocapsid transfer to many

surrounding cells near the attaching site via filopodia-like protrusions, although this hypothesis cannot fully explain large foci size at as early as 24 hpi. Alternatively, filopodia-like protrusions may not only penetrate the BL barrier but also degrade it in the neighboring area. Invadosomes in mammals are known to secrete enzymes to degrade the extracellular matrix. Considering our SEM observation of thin mesh-like structures spreading in a wide area from attached hemocytes (Fig. 5 of the revised manuscript), these ARIF-1-related structures may also secrete protein degradation enzymes to breach a wider area of the BL barrier and hence circulating BVs (remaining of injected ones) can establish tissue infection. Further analyses are needed to verify these hypotheses, which we think will be the future work of our research. We added the discussion mentioned above in the Discussion section (lines 485–500 of the revised manuscript).

Minor comments:

1. Due to the intrinsic autofluorescence of silkworm tissues, reliance on fluorescence imaging alone may be inadequate. It is suggested that the authors add quantitative data to demonstrate the differences between wild-type and mutant viruses' infection more effectively. Such as genome copies or GFP expression levels.

(Authors' reply)

Thank you for your suggestion. In our previous study, we quantitated viral gene expression at 1–4 dpi and found that its increase was significantly delayed in the trachea and brain (Fig. 5 of Kokusho et al., *J. Gen. Virol.*, 2015, doi:10.1099/vir.0.000130). We added some description about this point (lines 117–121 of the revised manuscript).

2. Why is the host hsp70 promoter used to drive the expression of the reporter gene? Have the authors confirmed whether mutation in Arif-1 affect the expression of the hsp70 promoter?

(Authors' reply)

Thank you for your comment. We use the *Drosophila melanogaster hsp70* promoter that works well for marker gene expression throughout viral infection stages in hemocytes (Hori et al., *J. Invertebr. Pathol.*, 2013, doi:10.1016/j.jip.2012.09.004). Both hspGFP and hspGFP-ARIFD showed high GFP expression enough to distinguish virus-infected cells (Figs. 1, 2, and Supplementary Fig. S11 of the revised manuscript). Thus, we think these viruses are useful tools for monitoring systemic spread of infection. We added some

descriptions in the Materials and Methods section (lines 920–924 of the revised manuscript).

3. The authors should provide replication and proliferation data for several Arif-1 mutant viruses to exclude the possibility that Arif-1 promotes systemic infection by participating in viral replication and proliferation.

(Authors' reply)

Thank you for your constructive suggestion. We previously reported that the deletion of ARIF-1 did not affect BV and OB production in cultured cells (Fig. 2B of Kokusho et al., *J. Gen. Virol.*, 2015, doi:10.1099/vir.0.000130). In this revision, we have also examined the BV production and virus genome replication of truncated ARIF-1 mutants. While no significant decrease was observed in virus genome replication (Supplementary Fig. S10A), some ARIF-GFP variants showed significantly increased or decreased BV production (Supplementary Fig. S10B). These results imply that the expression of truncated ARIF-1 might affect budding efficiency, but we think this does not matter when examining actin remodeling by ARIF-1 at the early stage of infection. We added some descriptions about this point in the Results section (lines 440–445 of the revised manuscript).

4. What does Fig. 5D aim to illustrate?

(Authors' reply)

Thank you for your comment. Fig. 5D aims to show the co-localization of ARIF-1 and GP64. Because GP64 is a viral peplomer protein, its localization at filopodia-like protrusions and invadosome-like structures implies BV budding from these sites. In this revision, we added many new results of higher importance; thus, we moved this figure to Supplementary Fig. S7.

Thank you again, reviewer #2, for your valuable and constructive comments.

Reviewer #3

We thank reviewer #3 for the valuable and helpful comments.

All changes are indicated by using the Track Changes function of Microsoft Word. Please see the Word file of our article to see the changes indicated below (the line numbers of the article pdf file available in the editorial system may differ from those of the original Word file).

Baculoviruses are important insect pathogens and have also been broadly used as expression vectors. Studies dating to the 1980s have examined the mechanisms underlying how baculoviruses systemically infect insects with evidence implicating both the host tracheal system and hemocytes in disseminating the budded form of baculoviruses to different tissues. This study advances results reported by these authors in 2015 (JGV) on an alphabaculovirus gene named actin rearrangement-inducing factor 1(arif-1). These earlier studies indicated loss of ARIF-1 in BmNPV slowed systemic infection of *Bombyx mori* larvae. This study used a GFP-expressing recombinant BmNPV together with or without a functional arif-1 gene to further advance understanding of ARIF-1 functions. The authors report that BmNPV infects *B. mori* immune cells (hemocytes), which causes ARIF-1 induced alterations to the cytoskeleton that promote the formation of filipodia and an invadosome-like structure that promotes the entry of budded virus through the basal lamina and into tracheal cells. The authors further present microscopy-based evidence indicating budded then moves from trachea to other tissues like the fat body.

My overall assessment is that the study presents new, information on how alphabaculoviruses systemically infect hosts that will be of interest to both virologists and immunologists that study arthropods. While prior studies emphasize entry of BVs from the gut to tracheoblasts, this study shows that BVs can also be transferred from hemocytes to tracheal epithelial cells. Together with another recently published study (Lauko et al. 2021 MBC), this paper will also likely be of interest to the broader cell biology community as it shows that baculovirus-induced invadosomes show features that are similar to the invadosomes some vertebrate pathogens produce. The paper is overall well organized, clearly written, and the results overall support main conclusions. I have only a small number of suggestions for revision.

(Authors' reply)

We thank this reviewer for their encouraging comments.

1. A general but important omission from the paper in my opinion is the growing literature indicating a substantial number of hemocytes in some insects are intrinsically sessile. Some sessile hemocytes proliferate and thus contribute to the population of cells in circulation and also can dynamically change the proportion of cells in circulation. Most of this literature has accrued from studies of *Drosophila* and mosquitoes with relatively little literature in Lepidoptera (like *B. mori*). Finding these studies is also easily done by just key wording 'sessile hemocytes AND insect' in a general literature search. Despite the bias of the current literature toward Diptera, there are strong reasons to suspect that sessile hemocytes are a common phenomenon in insects generally including moth larvae. This is important to this study for obvious reasons as it would suggest the possibility baculovirus infection alters how many or few sessile hemocytes are present as opposed to fundamentally changing hemocytes from being exclusively in circulation. I thus would strongly suggest the authors bring this possibility into the discussion section of the manuscript and at least cite a couple of papers indicating the types of hemocytes that have been identified to bind to the surface of different tissues in the absence of virus infection and possible implications for the results reported in this study.

(Authors' reply)

Thank you for your constructive comment. We added some discussions about sessile hemocytes and their involvement in baculovirus infection (lines 209–213 and 786–795 of the revised manuscript). Also, we counted the number of hemocyte attachments and found a > 20-fold increase in the wild-type virus (hspGFP)-infected trachea (Fig. 3C of the revised manuscript). 78.1% of attached hemocytes were GFP- & Hemocytin-positive, whereas 10.1% were GFP-positive but Hemocytin-negative (Fig. 3E of the revised manuscript); these results suggest that ARIF-1-mediated hemocyte attachment mainly occurs in granulocytes, but other hemocyte types also have the ability to do so. We added the discussion on this point in the Discussion section (lines 795–823 of the revised manuscript).

2. Along these lines, the authors mention the possible roles of *fgf* in hemocyte migration and how this might fit with BmNPV encoding an *fgf* gene that could be relevant to

interactions with trachea. That's fine but given the broader literature, I would be suspicious *B. mori* hemocytes preferentially bind to only trachea when very careful studies in other insects present evidence sessile, granulocyte-like cells bind to a number of other tissues that are also enveloped by the basal lamina. The authors make no mention of whether they restricted their microscopy studies to only looking at trachea or alternatively potentially observed hemocytes (whether infected or not) on the surface of other tissues. I would suggest at that the authors at least generally address this point as either a key future need or that the possibly did see hemocytes on other tissues but didn't pursue this in the current study.

I fully recognize the authors' interest in the tracheal system is driven primarily by the baculovirus literature and prior studies that implicate it in promoting systemic infection. On the other hand, given more recent evidence describing sessile hemocytes on the surface of several tissues in insects, I cannot intuitively see why ARIF-1's role in promoting the formation of filopodia that penetrate the basal lamina should necessarily be restricted to the tracheal system given most tissues in the hemocoel are also covered by the basal lamina. In turn, this could suggest infected hemocytes could use invadosomes to penetrate the basal lamina to infect other cell types besides tracheal epithelia. If the authors have results indicating a real bias that infected granulocytes preferentially associate themselves with trachea, they should report it. If they don't, they should again at acknowledge this is an unknown that needs to be examined in the future.

(Authors' reply)

Thank you for your comment. We are sorry that our manuscript might have been misleading. We do not think *B. mori* hemocytes preferentially attach to the trachea. We agree with your comment that virus-infected hemocytes may also attach to other tissues. In fact, hemocyte attachment was also observed in the fat body. However, it was quite difficult to distinguish whether hemocytes attached to the fat body or the trachea because the fat body is mostly covered with a thin tracheal network (please see bright-field images in Figs. 2A, B of the revised manuscript). Some preliminary experiments implied attachment on tissues other than the trachea and the fat body, but we have not yet performed a convincing number of observations (we will examine this in the future). We revised the manuscript to state which tissues we observed clearly and to discuss the points mentioned above in the Discussion section (lines 829–840 of the revised manuscript).

3. Fig. 3. I would think an important graph to show in addition to the number of pH3+ hemocytes (Fig. 3D) and total number of hemocytes in circulation (Fig. 3E) would be the proportion of these cells that were BmNPV infected as evidenced by detection of GFP. Moreover, are these hemocytes predominantly granulocytes, which is the only hemocyte type the authors mention in the study as being infected, or are similar proportions of other hemocyte types in circulation (plasmatocytes, spherule cells, oenocytoids) also BmNPV infected (which I would suspect). If the latter, a possible future direction would also be if ARIF-1 only induces the formation of invadosomes in infected granulocytes or cytoskeletal rearrangements result in other hemocyte types also producing these structures (at minimum plasmatocytes which are also a strongly adhesive in *B. mori* and other Lepidoptera).

(Authors' reply)

Thank you for your constructive comment. We newly examined the ratio of GFP-positive hemocytes attached to the trachea and found that over 85% were GFP-positive at 24 hpi (Fig. 3D of the revised manuscript). Our previous study showed that about 75% of hemocytes were GFP-positive in both hspGFP- and hspGFP-ARIFD-infected larvae at 24 hpi (Fig. 4 of Kokusho et al., *J. Gen. Virol.*, 2015, doi:10.1099/vir.0.000130), although the concentration of injected BVs was not the same as the present study. Taken together with the significant increase in the number of attached hemocytes of hspGFP compared to mock and hspGFP-ARIFD (Fig. 3C of the revised manuscript), these results suggest that the majority of the hemocytes are virus-infected, but only ARIF-1-expressing hemocytes have an increased ability to attach to the tissue surface.

Additionally, we examined the ratio of Hemocytin expression in attached hemocytes and found that over 85% were Hemocytin-positive (Fig. 3D of the revised manuscript), suggesting that attached hemocytes are mainly granulocytes. On the other hand, 13% were Hemocytin-negative but GFP-positive (Fig. 3E of the revised manuscript), implying attachment of virus-infected hemocytes also occurs in other hemocyte types. We previously examined the infection rate of each hemocyte type and found that BmNPV can infect all hemocyte types (granulocytes, spherulocytes, plasmatocytes, and oenocytoids), but plasmatocytes were more resistant to BmNPV infection (Hori et al., *J. Invertebr. Pathol.*, 2013, doi:10.1016/j.jip.2012.09.004). Thus, as you pointed out, ARIF-1-induced hemocyte attachment and resulting actin remodeling

presumably occur in other hemocyte types, but at present, we do not have enough evidence to disclose the details. This is because it is difficult to distinguish types of virus-infected attached hemocytes by morphology, and we do not have suitable antibodies for hemocyte type markers other than the anti-Hemocytin antibody.

We added discussions and future directions about these points in the Discussion section (lines 795–823 of the revised manuscript).

4. I fully understand why the authors conducted their experiments by injecting budded virus into the hemocoel as oral infection would create confounding conditions that could make interpretation of the data presented. Still, injecting virus directly into the hemocoel is also ‘non-natural’. Thus, it could be valuable/important for the authors to show or at least mention from other studies that per os infected *B. mori* also results in early infection of both hemocytes and tracheoblasts and that hemocytes projecting invadosomes into tracheal epithelial cells occurs. This could easily be looked at if the authors haven’t already done so, and I think should be shown to make clear the results reported occur after normal infection.

(Authors’ reply)

Thank you for your constructive suggestion. We newly performed an oral infection assay of the ARIF-GFP virus. We found that, similar to the case of intrahemocoelic infection, virus-infected hemocytes attached to the tracheal surface and formed filopodia-like protrusions and invadosome-like structures at 48 hpi (Supplementary Figs. S4A–B of the revised manuscript). The tracheal infection started at the BL-surrounded area at 72 hpi and then expanded at 96 hpi (Supplementary Fig. S4A of the revised manuscript). These observations suggest that the hemocyte-mediated BL breach mechanism is also important in the natural route of infection. We added descriptions about this point in the Results section (lines 273–281 of the revised manuscript).

Minor points

1. While the authors do a good job of reporting N values in the graphs presented, I didn’t get a clear sense of the number of larvae or samples the authors examined to generate the microscopy data, and in essence how common or rare the granulocytes in association with trachea were as shown in Fig. 3-5. Some mention of sample sizes in the Methods or elsewhere for this aspect of the study would be helpful.

(Authors' reply)

Thank you for your comment. We improved the original images and added new figures with the number of hemocytes, structures, or signals observed (Figs. 3C–E, 4G, 6F, and 7D–E of the revised manuscript). Also, we added the information about sample sizes in the figure legends and the Methods section (lines 1086, 1095, 1124–1130, and 1189–1241 of the revised manuscript).

2. As noted above, the paper is overall well written but I did see a few sentences/phrasing that was a bit awkward. On the first page of the Introduction, for example, the authors use the term 'open blood-vascular system' (line 37) but this isn't wording that insect immunologists normally use. Instead, correct phrasing would simply be 'an open circulatory system'. Line 38, should be changed to 'with a basal lamina' from 'with the basal lamina'. Line 46, insect-related should be revised to insect-associated. Select other pages have similar minor grammatical/phrasing errors. Potentially just rereading carefully would be enough to catch these and correct them.

(Authors' reply)

Thank you for your comment. We are sorry for these awkward sentences and phrasing due to our English translation skills. We used an English editing service for the first manuscript to correct grammatical and phrasing errors, but some might have remained uncorrected. We carefully reread the revised manuscript and corrected the errors (major ones are listed below). We would appreciate it if you could point out uncorrected errors to us.

(Line numbers indicates those in the revised version.)

Line 38: "open blood-vascular system" was modified to "an open circulatory system."

Line 39: "with a basal lamina" was modified to "with the basal lamina."

Line 47: "insect-related" was modified to "insect-associated."

Line 78: "does" was modified to "do."

Line 131: "coined" was modified to "termed."

Line 218: "floating hemocytes" was modified to "circulating hemocytes".

Line 454 etc.: "attachment on" was modified to "attaching to." (We applied many similar corrections.)

Line 827: "may enables" was modified to "may enable."

Thank you again, reviewer #3, for your valuable and constructive comments.

Reviewer #1

We thank reviewer #1 for the valuable and helpful comments.

Review comments: (related to my specific concerns of the initial review)

1-it is much easier to discern the large vs small foci, and the quantification (Fig2E) really supports the claim.

2-the hemocyte staining is much better and is now quite convincing

(Authors' reply)

We appreciate this reviewer's encouraging comments.

3-the adjusted graph scaling looks good. However in Fig 8 it is still a bit deceiving showing mutant viral OB levels near zero, but WT OBs at 10^9 . Since authors do indeed state in the results that it's a 30-fold reduction for the mutants, it makes it clear the difference but also highlights the issue I have with the zero straight to 10^9 . It's a minor thing and I don't think it needs any adjustment (I have the same comment for the graph in fig 3F).

(Authors' reply)

Thank you for your constructive comment. To improve the deceiving appearances of Fig. 8C–D, Fig. 3C (I think you probably intended to mention this figure, not Fig. 3F) and additionally Fig. 1C, we changed the Y-axes to two-segmented ones so that we can see the exact value of each sample more easily.

4-the microscopy images are well improved, and the additional immunoblot analyses are a nice addition...but they do show there is much less of the dC1 and dC2 proteins (and a lot more of the dC3 protein), perhaps indicating maturation problems, but I do agree there is enough detectable in the immunostained cells.

(Authors' reply)

Thank you for your agreement that the expression of these proteins was enough to detect in the immunostained cells.

5-The terminal carcass OB levels, in each gram weight of tissue, is much more convincing data.

(Authors' reply)

Thank you for your comment. The terminal carcass OB levels per gram weight of the cadaver is shown in Supplementary Fig. S11A.

An outstanding issue to resolve:

One issue that needs to be confirmed/addressed is that the authors state (in fig 2E, lines 146 and 651) that the size of the “large” infection foci is 1000 mm². When I did a quick online conversion of area units, it equated 1000 mm² to about 1.5 inches², and so I think this sizing may be at least an order of magnitude off (more like 100 mm²?).

(Authors’ reply)

Thank you for your comment. We sincerely apologize for the mistake in the unit. The correct unit is 1000 μm³, not mm³. We corrected them in the Results section (line 208), the Methods section (line 805), Fig. 2D, and the legend of Fig. 2E (line 1402).

Overall, I think this is a very interesting and convincing set of experiments and that the authors’ statements reflect the data well.

(Authors’ reply)

We deeply appreciate this reviewer’s encouraging comments.

Reviewer #2

We thank reviewer #2 for the valuable and helpful comments.

The author has thoroughly addressed the issues I raised in current revised manuscript. In particular, the scanning electron microscopy results and the findings regarding filopodia-like protrusions and vp39 localization have significantly improved the completeness and credibility of this work. This study will greatly advance our understanding of the mechanisms underlying the rapid spread of baculoviruses within the host. I have no additional comments.

(Authors' reply)

We sincerely thank you once again for providing us with constructive comments and suggestions, which have significantly contributed to enhancing the quality and reliability of our manuscript.

Reviewer #3

We thank reviewer #3 for the valuable and helpful comments.

I was Reviewer #3 of the first submission. I read the revised manuscript and thought the authors did an excellent job addressing my prior concerns. It also appeared they did a very good job of carefully and thoughtfully addressing the concerns of the other two reviewers.

(Authors' reply)

We deeply appreciate this reviewer's encouraging comments.

In my previous review, I also pointed out a handful of grammatical/syntax errors. The authors corrected these and asked that I specify any other suggested corrections I might see. I have done this below. I did not previously mention this, but I also suggest the authors be a little more precise about arif-1 only being found in certain baculoviruses (not all), which is another feature that is likely important for function.

(Authors' reply)

We sincerely appreciate your assistance for improving our English sentences. We provide a point-by-point response to the revisions you suggested.

Abstract and Introduction--host's body fluid: host hemocoel (or host's circulatory system)

(Authors' reply)

Thank you for your comment. We have revised the wording according to your suggestion (line 13 in the Abstract, lines 40 and 43 in the Introduction).

Introduction

Baculovirus is: Should be Baculoviruses are

(Authors' reply)

Thank you for your comment. We have revised the wording according to your suggestion (line 53).

Next, baculoviruses need to cross the midgut BL to establish infection in other tissues. Correct to 'Next, most baculoviruses cross the midgut BL...'

(Authors' reply)

Thank you for your comment. We have revised the wording according to your suggestion (line 64).

I suggest this because some baculoviruses only infect the midgut of hosts and do not establish systemic infections. The authors should also state either in the Introduction, Discussion or both that *arif-1* isn't a baculovirus core gene. I did not take the time to look carefully myself, but George Rohrmann's 2019 review reports that *arif-1* is present in most/all alphabaculoviruses (Group I and II) but is apparently absent in betabaculoviruses (GVs) (which also systemically infect hosts), or gamma- and deltabaculoviruses (which primarily or exclusively infect only the midgut). Thus, my overall suggestion is for the authors not use the term 'baculovirus' when referring to *arif-1* and more precisely refer to its presence in alphabaculoviruses (=nucleopolyhedroviruses (NPVs) which infect Lepidoptera).

(Authors' reply)

Thank you for your constructive suggestion. We have revised the whole manuscript to achieve more precise descriptions (lines 103–105, 174, 475, 543, 568) and added the information of four baculovirus genera (lines 55–59) and *arif-1*'s conservation (lines 158–159).

Alternatively, baculoviruses utilize the host trachea as an escape route from the midgut (63 Supplementary Fig. S1A, [iii-b]) (14). Revise slightly by adding the word 'most' before baculoviruses' (for the same reason as noted above).

(Authors' reply)

Thank you for your comment. We have revised the wording according to your suggestion (lines 103–105).

In this study, we discovered that the baculovirus membrane protein ARIF-1 (actin rearrangement-inducing factor 1) modifies the actin cytoskeleton of host hemocytes and enables them to breach the BL barrier, establishing a highly efficient systemic infection.

Modify for precision to: In this study, we discovered that the membrane protein ARIF-1 (actin rearrangement-inducing factor 1) of *Bombyx mori* nucleopolyhedrovirus modifies

the actin cytoskeleton of host hemocytes and enables them to breach the BL barrier, establishing a highly efficient systemic infection.

(Authors' reply)

Thank you for your comment. We have revised the wording according to your suggestion (lines 124–125).

Results

First subheading. Baculovirus can directly establish infection from the hemolymph at the central tracheal region despite the BL barrier'

Would suggest the authors revise this subheading to 'Bombyx mori nucleopolyhedrovirus can directly establish infection from the hemolymph at the central tracheal region despite the BL barrier' (again, because not all baculoviruses systemically infect hosts).

(Authors' reply)

Thank you for your comment. We have revised the wording according to your suggestion (lines 129, 180, 249).

Line 104. Change viral arif-1 gene to BmNPV arif-1 gene.

(Authors' reply)

Thank you for your comment. We have revised the wording according to your suggestion (line 156).

Line 159. Baculovirus-infected hemocytes attach to the invasive infection foci in the trachea

Revise to: BmNPV- infected hemocytes attach to the invasive infection foci in the trachea

(Authors' reply)

Thank you for your comment. We have revised the wording according to your suggestion (line 221).

Line 181. disappeared from the hemolymph by BmNPV infection. Revise to 'disappeared from the hemolymph after BmNPV infection'.

(Authors' reply)

Thank you for your comment. We have revised the wording according to your suggestion (line 245).

Line 321-325. In summary, our results suggest that baculoviruses remodel host hemocytes to form ARIF-1-derived structures for attaching to the tissue surface and then breach the BL barrier to deliver progeny virions directly into the tissues. This ARIF-1-mediated hemocyte remodeling greatly contributes to the quick spread of virus infection throughout the host body.

Revise to: In summary, our results suggest that BmNPV remodels host hemocytes to form ARIF-1-derived structures for attaching to the tissue surface and then breach the BL barrier to deliver progeny virions directly into the tissues. This ARIF-1-mediated hemocyte remodeling greatly contributes to the quick spread of virus infection throughout the hos.

(Authors' reply)

Thank you for your comment. We have revised the wording according to your suggestion (line 396).

Discussion

ARIF-1 has been reported as a baculoviral inducer of actin rearrangement in lepidopteran cultured cells (17).

Revise to: ARIF-1 has been reported as an inducer of actin rearrangement in lepidopteran cultured cells after BmNPV infection (17).

(Authors' reply)

Thank you for your comment. We have revised the wording to “ARIF-1 has been reported as an inducer of actin rearrangement in lepidopteran cultured cells after alphabaculovirus infection¹⁸” (lines 374–375), because AcMNPV is used in reference #17 (#18 of the revised manuscript).

Line 506. Since baculoviruses highly express arif-1 during the initial stage of midgut infection (71), ARIF-1-mediated cytoskeletal modification may also occur in the midgut.

Modify to ‘Since baculoviruses like BmNPV highly express arif-1 during the initial stage of midgut infection (71), ARIF-1-mediated cytoskeletal modification may also occur in the midgut.

(Authors’ reply)

Thank you for your comment. As reference #71 (#74 of the revised manuscript) reports the transcriptome analysis of AcMNPV-infected *T. ni* midgut, we have revised the wording to “Since AcMNPV highly expresses *arif-1* ...” (line 637–638).

Currently, one plausible theory of the midgut escape is that baculoviruses overcome the midgut BL barrier by escaping to the hemocoel through tracheoblast infection (14), as described in the Introduction section and Supplementary Fig. S1.

Again, suggest revision to: ‘Currently, one plausible theory of the midgut escape is that BmNPV and other alphabaculoviruses like AcMNPV overcome the midgut BL barrier by escaping to the hemocoel through tracheoblast infection (14), as described in the Introduction section and Supplementary Fig. S1.

Following up on this, the authors maybe should also add a followup sentence indicating the above 'plausible' theory potentially applies to most or all alphabaculoviruses (NPVs) but not other baculoviruses given the apparent absence of arif-1 genes in betabaculoviruses which also escape the midgut. On the other hand the absence of arif-1 in delta and gammabaculviruses potentially contributes to viruses in these genera infecting primarily or exclusively the midgut of hosts.

(Authors’ reply)

Thank you for your comment. We speculate that the “plausible theory” (the tracheoblast-mediated method) is the common midgut escape strategy among baculoviruses with midgut escape ability, and the ARIF-1-mediated midgut escape might be the additional method acquired in alphabaculoviruses because *arif-1* is absent in beta-, gamma-, and deltabaculoviruses and some Group II alphabaculoviruses (as you pointed out). Our manuscript might be confusing, so we revised the last paragraph in the Discussion section (lines 627–657).